 **eLIFE**

# Viral miRNA adaptor differentially recruits miRNAs to target mRNAs through alternative base-pairing

Carlos Gorbea[1], Tim Mosbruger[2], David A Nix[3], Demián Cazalla[1]*

[1]Department of Biochemistry, University of Utah School of Medicine, Salt Lake City, United States; [2]Children's Hospital of Philadelphia, Philadelphia, United States; [3]Huntsman Cancer Institute, University of Utah School of Medicine, Salt Lake City, United States

**Abstract** HSUR2 is a viral non-coding RNA (ncRNA) that functions as a microRNA (miRNA) adaptor. HSUR2 inhibits apoptosis in infected cells by recruiting host miRNAs miR-142–3p and miR-16 to mRNAs encoding apoptotic factors. HSUR2's target recognition mechanism is not understood. It is also unknown why HSUR2 utilizes miR-16 to downregulate only a subset of transcripts. We developed a general method for *i*ndividual-nucleotide resolution *R*NA-RNA *i*nteraction identification by *c*rosslinking and *c*apture (iRICC) to identify sequences mediating interactions between HSUR2 and target mRNAs in vivo. Mutational analyses confirmed identified HSUR2-mRNA interactions and validated iRICC as a method that confidently determines sequences mediating RNA-RNA interactions in vivo. We show that HSUR2 does not display a 'seed' region to base-pair with most target mRNAs, but instead uses different regions to interact with different transcripts. We further demonstrate that this versatile mode of interaction via variable base-pairing provides HSUR2 with a mechanism for differential miRNA recruitment.
DOI: https://doi.org/10.7554/eLife.50530.001

*For correspondence:
dcazalla@biochem.utah.edu

Competing interests: The authors declare that no competing interests exist.

## Introduction

Herpesviruses can establish lifelong latent infections. During latency, herpesviruses employ different classes of ncRNAs (*Tycowski et al., 2015*) to control gene expression and modulate processes such as apoptosis and the immune response. *Herpesvirus saimiri* (HVS) is an oncogenic γ-herpesvirus that establishes latent infections in T cells of New World monkeys (*Ensser and Fleckenstein, 2005*) and expresses seven small nuclear RNAs (snRNAs) of the Sm-class called HSURs (*Albrecht and Fleckenstein, 1992*; *Lee et al., 1988*; *Murthy et al., 1986*; *Wassarman et al., 1989*) during latency.

So far, functions have only been assigned to HSUR1 and HSUR2. These two viral snRNAs bind host miRNAs (*Cazalla et al., 2010*). HSUR1 binds with perfect complementarity to the seed region of miR-142–3p, whereas it base-pairs with the seed region and the 3' end of miR-27. Interaction of HSUR1 and miR-27 results in the degradation of this miRNA by an unknown mechanism (*Cazalla et al., 2010*; *Pawlica et al., 2016*). HSUR2 contains sequences that base-pair with perfect complementarity with the seed regions of miR-142–3p and miR-16 without affecting the abundance or activity of these miRNAs. Rather, HSUR2 functions as a miRNA adaptor that base-pairs with host mRNAs and recruits miR-142–3p and miR-16 to promote destabilization of the target mRNAs (*Gorbea et al., 2017*). HVS utilizes this mechanism to downregulate expression of several proteins including apoptotic factors in latently infected T cells (*Gorbea et al., 2017*). Psoralen-dependent crosslinking suggests that HSUR2 base-pairs with target mRNAs (*Gorbea et al., 2017*), but it has not been determined if base-pairing is required for HSUR2-mediated mRNA repression. Other questions regarding the mechanism and specificity of HSUR2-mediated mRNA repression remain

unanswered. Most importantly, the regions of target mRNAs bound by HSUR2 and the sequences mediating such interactions are not known. It is also unclear why the interaction between HSUR2 and miR-16 is required for destabilization of only a subset of, but not all, target mRNAs (*Gorbea et al., 2017*).

Small ncRNAs that function as guides for the recruitment of effector complexes to target RNAs usually use a specific subregion, or seed sequence to interact with the target transcript. Small nucleolar RNAs (snoRNAs) (*Decatur and Fournier, 2003*), miRNAs (*Bartel, 2018*), CRISPR RNAs (crRNAs), and bacterial Hfq-dependent small RNAs (*Gorski et al., 2017*) provide examples of different systems in which the folding of the guide RNA into secondary or tertiary structures exposes a defined sequence that is used for binding to the target RNAs. Complementarity between this discrete seed sequence and the target RNA is sufficient to confer specificity to the interaction. Members of the Sm-class of ncRNAs part from this norm. In some cases, they can use a subregion to interact with rather diverse sequences through distinct arrangements of base pairs. For example, the 5' end of U7 snRNA can interact with a series of similar sequences downstream of histone genes to promote the 3' end processing of histone pre-mRNAs (*Cotten et al., 1988*), whereas the 5' end of U1 snRNA interacts, also through flexible base-pairing, with a highly diverse sequence element that defines the 5' end of introns (*Roca and Krainer, 2009*). Moreover, Sm-class RNAs can use different subregions to interact with different RNA-binding partners. For example, U2 snRNA utilizes two different subregions to interact with the branch point sequence and U6 snRNA (*Nilsen, 1994*; *Staley and Guthrie, 1998*). This characteristic flexibility of Sm-class RNAs in both the use of subregions for interactions with target RNA transcripts and in the arrangement of base pairs undercuts efforts to predict potential binding sites for HSUR2 in target mRNAs.

To understand how HSUR2 interacts with its target mRNAs, we developed a method called iRICC to determine sequences mediating RNA-RNA interactions in vivo for an RNA of interest. iRICC was able to reveal sequences mediating interactions between HSUR2 and 171 binding sites in 110 target mRNAs without any previous knowledge of the regions of target mRNAs bound by HSUR2 or the binding properties of this viral ncRNA. HSUR2 does not display a specialized or seed sequence to bind most target mRNAs. Similar to U2 snRNA, HSUR2 can use different subregions to interact with different target mRNAs. Just like U1 or U7 snRNAs, HSUR2 also allows for diverse arrangements of base pairs in its interactions with target mRNAs. Mutational analyses confirmed the sequences mediating HSUR2-mRNA interactions, establishing iRICC as a robust method that can be broadly applied to identify functional RNA-RNA interactions in vivo. Finally, we show that base-pairing between HSUR2 and the target mRNA determines the differential recruitment of miR-16 to target mRNAs and its use in HSUR2-mediated mRNA destabilization.

## Results

### iRICC identifies sequences mediating RNA-RNA interactions in vivo

We had previously developed a method named RICC (*R*NA-RNA *i*nteraction identification by *c*rosslinking and *c*apture) to identify mRNAs that interacted with HSUR2 in virally infected cells (*Gorbea et al., 2017*). This method allowed for the identification of HSUR2 target mRNAs, but did not provide any information about the exact site of interaction, or the sequences mediating such interactions. In this study, we further modified the RICC protocol to develop a method we call iRICC that determines the sequences mediating RNA-RNA interactions in vivo through the identification of the actual psoralen-crosslinked residues.

HVS-transformed marmoset T cells were irradiated with long-wave (365 nm) UV light in the presence of the psoralen derivative 4'-aminomethyltrioxalen (AMT) to crosslink RNA duplexes in vivo (*Calvet and Pederson, 1979*; *Cimino et al., 1985*). Polyadenylated (polyA+) RNA was selected and hybridized to an antisense DNA oligonucleotide complementary to full-length HSUR2 (anti-H2 DNA) followed by limited single-strand S1 endonuclease digestion to fragment polyA+ RNA while maintaining HSUR2 integrity (*Figure 1A* and *Figure 1—figure supplement 1A*). After removal of anti-H2 DNA by DNase I digestion, samples were split into two samples and hybridization with anti-H2 DNA was repeated in one (Control) sample, followed by digestion of both samples with RNase H, which resulted in removal of HSUR2 only from the Control sample (*Figure 1A* and *Figure 1—figure supplement 1A*). Following a second digestion with DNase I, HSUR2 and crosslinked RNAs were

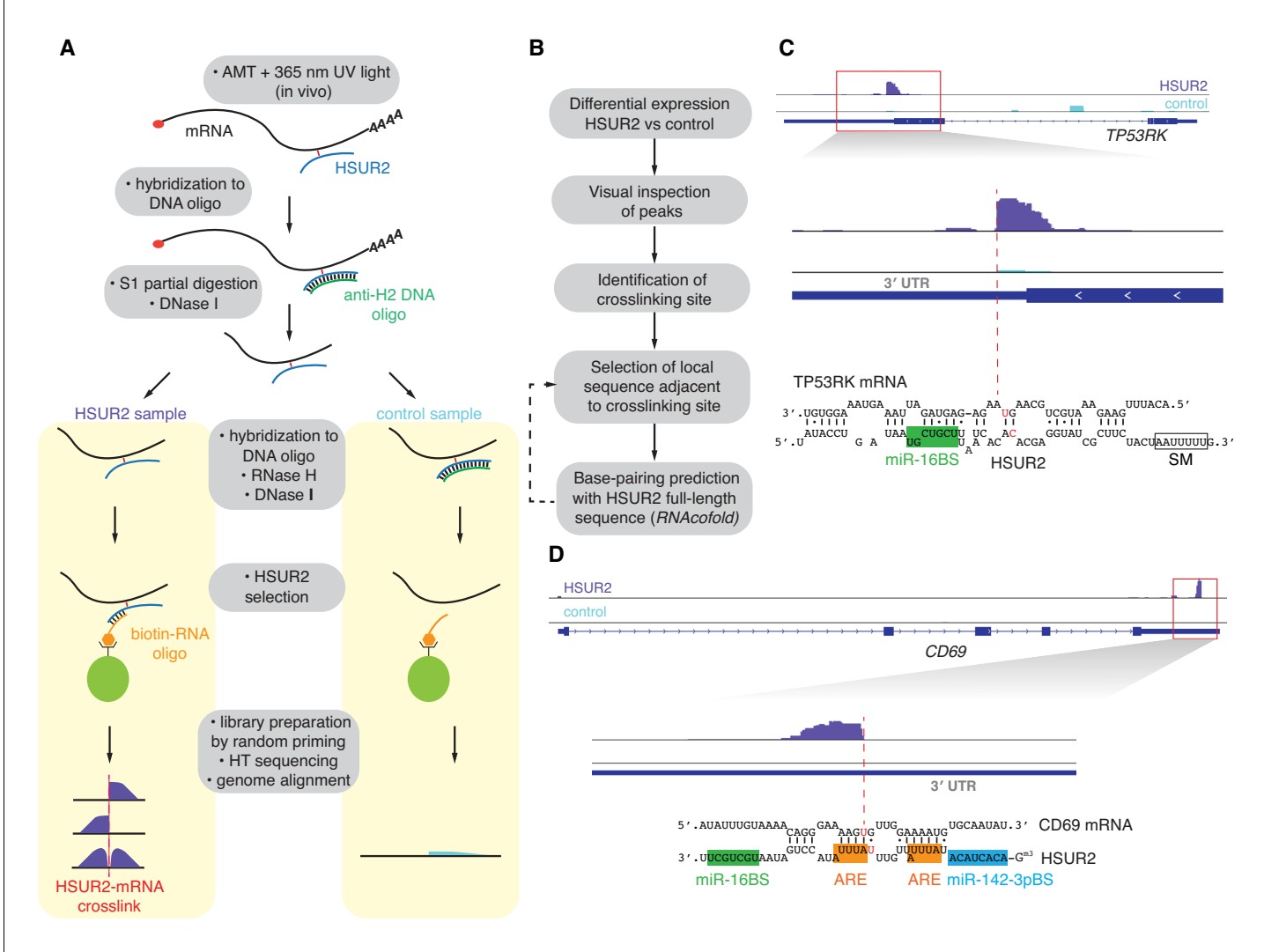

**Figure 1.** iRICC defines sequences that mediate HSUR2-mRNA interactions. (**A**) Schematic overview of the iRICC protocol. Critical steps: in vivo ATM crosslinking, protection of HSUR2 and partial RNA digestion with S1 nuclease, depletion of HSUR2 in control sample by RNase H digestion, HSUR2 pulldown, and strand-specific library preparation for high-throughput sequencing. Since crosslinking is not reversed, reverse transcriptase stalls or starts around the site of crosslinking. A sharp drop of aligned reads results in peaks with abrupt edges indicating the site of crosslinking between HSUR2 and target RNAs. (**B**) Outline of analysis to determine sequences involved in interactions between HSUR2 and target RNAs. (**C**) Zoomed-in view to the *TP53RK* gene for HSUR2 (purple) and Control (Cyan) samples. Dashed red line denotes abrupt drop of aligned reads indicating the site of crosslinking. Predicted base-pairing between HSUR2 and *TP53RK* sequences adjacent to the site of crosslinking is shown. Putative psoralen-crosslinked nucleotides are shown in red. Binding sites for miR-16 (green box) and Sm proteins (empty box) are indicated. One representative experiment (out of three performed) is shown. (**D**) Same as in (**C**) for *CD69* gene showing the ARE-like sequence (orange boxes) and binding site for miR-142–3p (blue box) in HSUR2.

DOI: https://doi.org/10.7554/eLife.50530.002

The following source data and figure supplements are available for figure 1:

**Figure supplement 1.** iRICC identifies HSUR2 target mRNAs.
DOI: https://doi.org/10.7554/eLife.50530.003
**Figure supplement 1—source data 1.** Source data for *Figure 1—figure supplement 1*.
DOI: https://doi.org/10.7554/eLife.50530.004
**Figure supplement 2.** iRICC defines sequences involved in HSUR2-mRNA interactions.
DOI: https://doi.org/10.7554/eLife.50530.005

purified from both samples under stringent conditions with a biotinylated RNA oligonucleotide antisense to HSUR2. High-throughput sequencing libraries were prepared from both samples using random primers. HSUR2 target mRNAs were identified by comparison of pulldowns from HSUR2 samples with Control samples in three independent replicate experiments (*Figure 1B* and *Supplementary file 1*). HSUR2 samples were highly reproducible across biological replicates ($R$ = 0.91–0.94). Gene Ontology (GO)-term enrichment analysis revealed that HSUR2 regulate expression of genes that encode proteins with roles in the regulation of apoptosis and cell cycle progression, processes that are relevant in herpesviral latency (*Figure 2—figure supplement 1D* and *Supplementary file 4*). Targets include previously identified HSUR2 target mRNAs (e.g. NGDN) (*Gorbea et al., 2017*) and additional mRNAs encoding pro-apoptotic factors (e.g. caspase 3, STK4, IRF1 and Rassf1; *Supplementary file 1*). Transient knockdown of HSUR2 expression with an antisense oligonucleotide (HSUR2 ASO) (*Cazalla et al., 2010*; *Gorbea et al., 2017*) confirmed that iRICC identified functional targets of HSUR2 (*Figure 1—figure supplement 1B*).

Regions of base-pairing between HSUR2 and target mRNAs were identified by visual examination of aligned reads. Reverse transcriptase elongation is blocked by psoralen-mediated crosslinking (*Ericson and Wollenzien, 1988*) of intramolecular and intermolecular RNA-RNA interactions, which results in a steep drop of aligned reads at the site of crosslinking (*Figure 1C,D* and *Figure 1—figure supplement 2A–C*). Potential base-pairing between HSUR2 and the local target sequence at the site of crosslinking was examined using the RNAcofold program to search for RNA-RNA interactions considering internal structures in both RNAs (*Bernhart et al., 2006*) (*Figure 1B*). Base-pairing between HSUR2 and regions of target mRNAs was indicated by a steep drop of reads in most (78%) of the cases, or by the edge of the peak in some (16.5%) cases (*Figure 1—figure supplement 2D*). In a few cases (5.5%), a sharp drop of aligned reads occurred at mRNA regions that did not show significant complementarity with HSUR2 and were therefore not considered for further analysis. iRICC identified and determined the sequences comprising 171 binding sites for HSUR2 in 110 target mRNAs (*Supplementary file 1* and *Supplementary file 2*). Psoralen is most reactive with juxtaposed pyrimidines at the end of helical runs or at G-U wobbles (*Bachellerie and Hearst, 1982*; *Thompson et al., 1982*). This arrangement of pyrimidines was present within 10 nucleotides of the position marked by the sharp drop of aligned reads in ~89% of the HSUR2-mRNA duplexes identified (*Figure 1—figure supplement 2E*), supporting the base-pairing predicted by RNAcofold.

## HSUR2 binding properties

HSUR2-binding site distribution on mRNAs is reminiscent of that of miRNAs (*Chi et al., 2009*; *Hafner et al., 2010*). HSUR2 binding sites were enriched in the 3′UTRs of target mRNAs (~67%, *Figure 2A*), with a slight preference for positions proximal to stop codons (*Figure 2—figure supplement 1A*). A substantial (~26%) fraction of HSUR2 binding sites were located in the coding sequence of target mRNAs, a location where miRNAs can exert regulation (*Fang and Rajewsky, 2011*; *Schnall-Levin et al., 2010*), and a smaller fraction (~6%) of binding sites overlapped with the stop codon. Only a small number (~2%) of binding sites were located in 5′UTRs. Most HSUR2 target mRNAs (~58%) contain one HSUR2 binding site, with smaller subsets of target mRNAs containing up to five bindings sites (*Figure 2B*), suggesting the possibility that, like miRNAs, several molecules of HSUR2 can cooperatively repress specific target mRNAs (*Grimson et al., 2007*; *Saetrom et al., 2007*).

## Functional validation of HSUR2-mRNA interactions

To functionally validate interacting sequences in HSUR2 and target mRNAs identified by iRICC, we utilized luciferase reporter gene assays (*Figure 3A*). HSUR2 did not downregulate the expression of firefly luciferase transcripts that lacked a 3′UTR or were fused to the 3′UTR of BCCIP, a gene not targeted by HSUR2 (*Figure 1—figure supplement 1B*), arguing that the firefly luciferase transcript does not contain functional HSUR2 binding sites (*Figure 3B*). In contrast, HSUR2 repressed to varying extents the expression of all six tested reporter genes carrying partial 3′UTRs of target mRNAs containing single HSUR2 binding sites identified by iRICC (*Figure 3B*). No repression of luciferase transcripts was observed when a mutant version of HSUR2 that cannot bind miR-142–3p (HSUR2Δ142–3p) and does not repress target mRNAs (*Gorbea et al., 2017*) was expressed

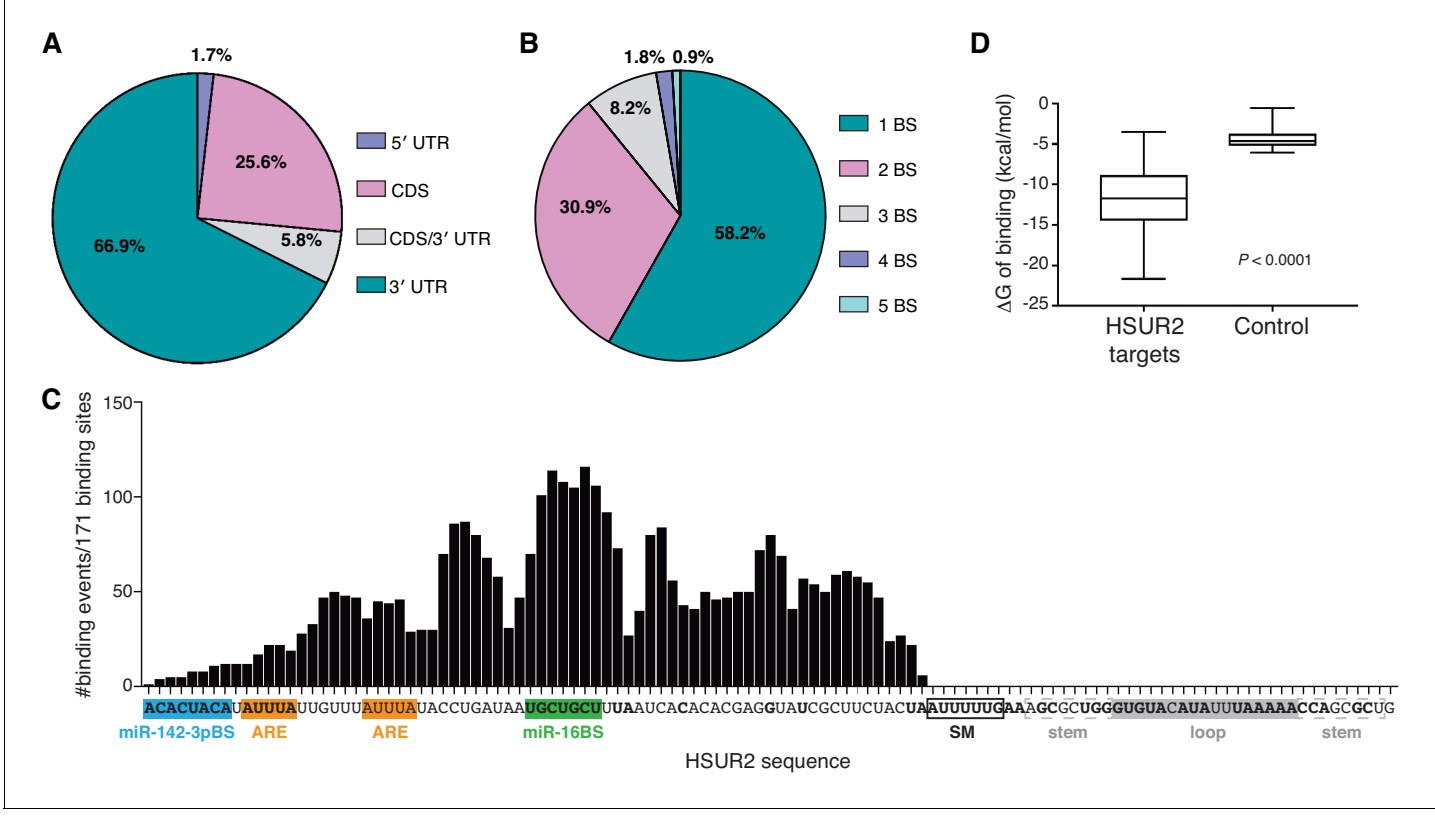

**Figure 2.** HSUR2 binding properties. (A–B) Location (A) and number (B) of HSUR2 binding sites in mRNAs (n = 171). (C) HSUR2 does not display a seed sequence that is involved in all interactions with target mRNAs. The graph shows the number of times that each HSUR2 nucleotide is involved in base-pairing with a target mRNA (171 HSUR2-mRNA interactions analyzed, see **Supplementary file 2**). Bold nucleotides are conserved among the different strains of HVS and also in *H. ateles* (**Cazalla et al., 2010**). ARE-like sequences, miRNA binding sites, and SM binding sequence are shown. (D) Average free energy of binding (ΔG) of interactions between HSUR2 and HSUR2-binding sites (HSUR2 targets, n = 171) or with 250 length-matched, randomly selected 3'UTR sequences (Control).

DOI: https://doi.org/10.7554/eLife.50530.006

The following source data and figure supplements are available for figure 2:

**Source data 1.** Source data for *Figure 2*.
DOI: https://doi.org/10.7554/eLife.50530.009

**Figure supplement 1.** Analysis of HSUR2 binding sites.
DOI: https://doi.org/10.7554/eLife.50530.007

**Figure supplement 1—source data 1.** Source data for *Figure 2—figure supplement 1*.
DOI: https://doi.org/10.7554/eLife.50530.008

(*Figure 3B*), indicating that repression of reporter transcripts occurs through HSUR2-mediated miRNA recruitment.

To test whether interactions between HSUR2 and target mRNAs are mediated by sequences identified by iRICC we performed mutational analyses. In every case, HSUR2 did not affect the expression of mutant reporter genes carrying 3'UTRs with point mutations predicted to disrupt interactions with HSUR2 (*Figure 4* and *Figure 4—figure supplement 1*), indicating that iRICC identified functional HSUR2 binding sites. Mutant versions of HSUR2 (*Figure 6—figure supplement 1*) with compensatory mutations predicted to restore complementarity with the corresponding mutant 3'UTR in all cases showed similar repressive activity when compared to wild-type HSUR2 (*Figure 4* and *Figure 4—figure supplement 1*). Collectively, these results confirm that base-pairing between HSUR2 and target mRNAs is required for HSUR2-mediated mRNA repression, and that the interactions are mediated by the sequences identified by iRICC.

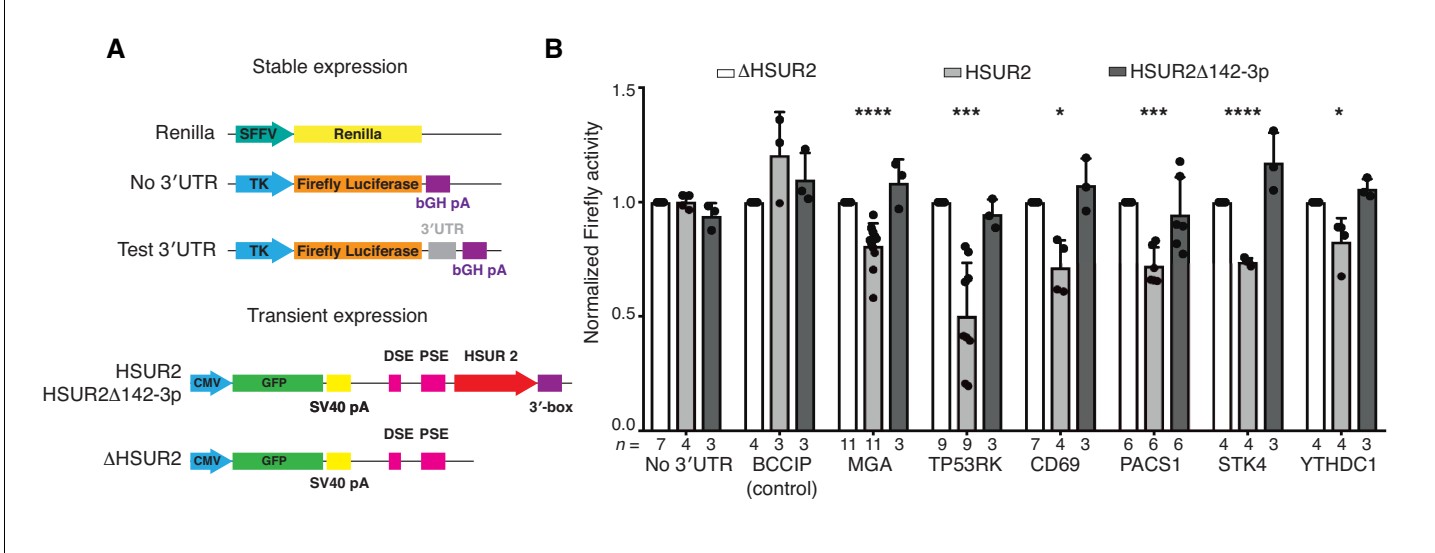

**Figure 3.** iRICC-seq identifies functional HSUR2 binding sites. (**A**) Schematic representation of plasmid constructs used for stable expression of luciferase reporters and transient expression of HSUR2. U937 cells were sequentially transduced with lentiviruses carrying the *Renilla* luciferase gene under the spleen focus-forming virus (SFFV) promoter and Firefly luciferase gene driven by the thymidine kinase (TK) promoter with no 3′UTR, full-length 3′UTR of BCCIP (control) or partial 3′UTR of HSUR2-target mRNAs containing single binding sites identified by iRICC-seq. Plasmids carrying GFP and either HSUR2 promoter alone (ΔHSUR2), wild-type HSUR2 (HSUR2), or a mutant version of HSUR2 that cannot bind miR-142–3p and does not promote mRNA destabilization (HSUR2Δ142–3p) were used for transient transfection. bGH: bovine growth hormone; PSE: proximal sequence element; DSE: distal sequence element. (**B**) Luciferase assays confirm binding sites for HSUR2. Luciferase-expressing U937 cells were transiently transfected with the indicated plasmids. Dots represent mean values of independent experiments with error bars representing s.d. Two-sided one sample Student's t-test vs. control (ΔHSUR2) set at 1.0. Sample size (n) is indicated below the chart in each case. *p<0.05, ***p<0.001, ****p<0.0001.
DOI: https://doi.org/10.7554/eLife.50530.010

The following source data is available for figure 3:

**Source data 1.** Source data for *Figure 3*.
DOI: https://doi.org/10.7554/eLife.50530.011

## HSUR2 binds target mRNAs through a flexible arrangement of base-pairs

Analysis of HSUR2 binding sites in target mRNAs suggested that HSUR2 does not display a seed region to interact with most target mRNAs (*Supplementary file 2*). HSUR2 not only uses different regions to interact with different target mRNAs, but it also uses variable arrangement of base pairs to mediate these interactions (*Figure 4* and *Supplementary file 2*). We hypothesized that the thermodynamic stability of the interaction with the target mRNA, rather than a specific arrangement of base pairs, determines the ability of HSUR2 to repress interacting mRNAs. To test this hypothesis, we generated PACS1 and TP53RK 3′UTR reporter constructs carrying point mutations in HSUR2 binding sites that result in interactions through different base-pairing arrangements, while maintaining similar binding stabilities (*Figure 5*). HSUR2 could repress both mutant reporters as efficiently as wild-type reporters (*Figure 3B* and *Figure 5*), indicating that HSUR2 can still bind to the modified sequences. Mutagenesis of HSUR2 indicated that interactions between wild-type HSUR2 and these mutant reporter transcripts occurred at the new designed sites (*Figure 5*). These observations indicate that the stability of the interaction between HSUR2 and its target mRNA, rather than any specific base-pairing arrangement, dictates HSUR2-mediated mRNA repression.

## HSUR2-mRNA base-pairing determines when miR-16 is needed for repression

Interaction between HSUR2 and miR-16 is required for repression of only a subset of target mRNAs (*Gorbea et al., 2017*). We noticed that the miR-16 binding site in HSUR2 (miR-16BS, *Figure 1C*) is directly engaged with the target in ~80% of mRNA interactions (*Figure 2—figure supplement 1C*), and hypothesized that once the miR-16BS is engaged in base-pairing with a target mRNA, HSUR2

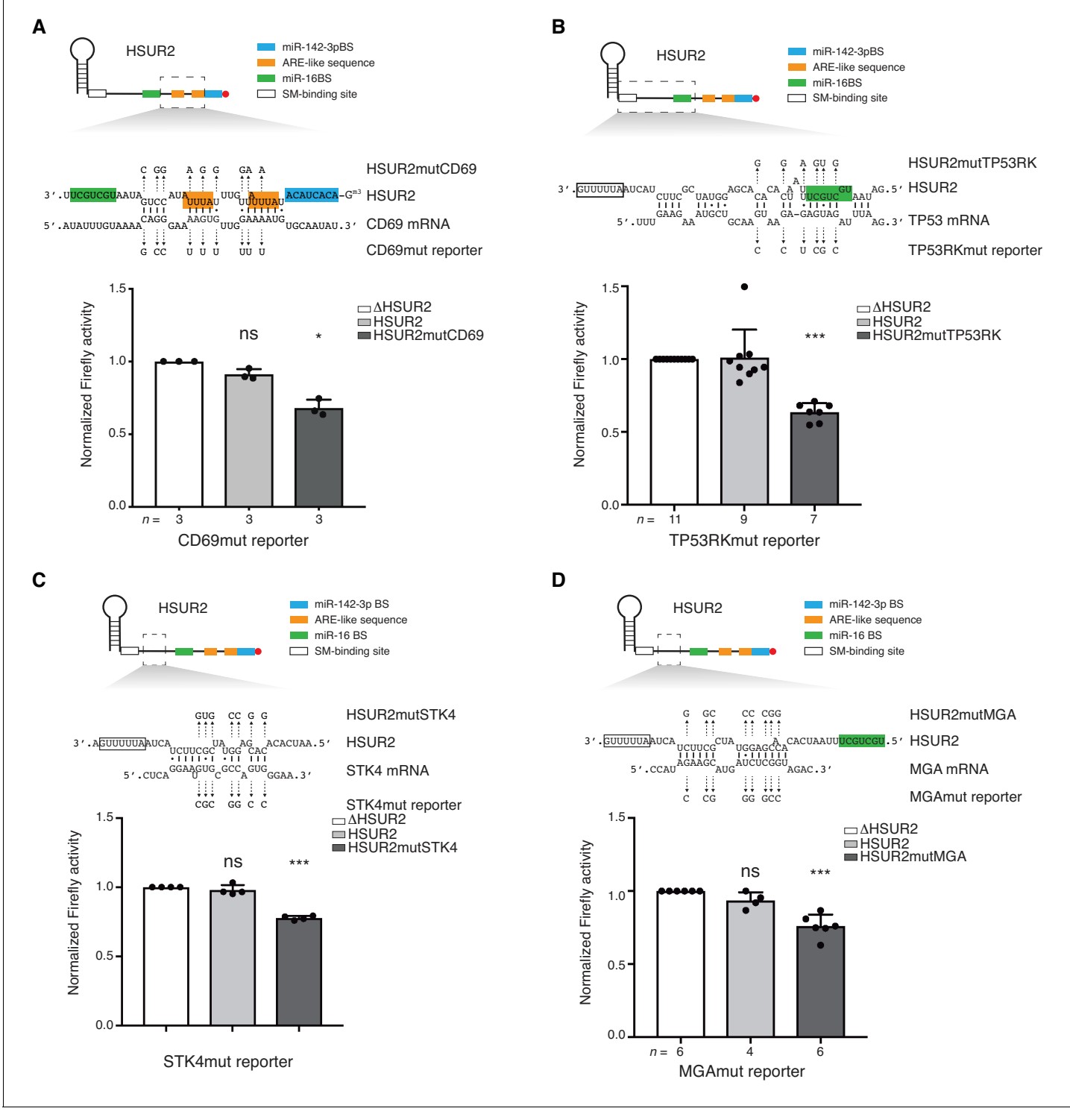

**Figure 4.** Base-pairing is required for HSUR2-mediated mRNA regulation. (**A**) Same as in *Figure 3B* but with cells stably expressing Firefly luciferase followed by partial CD69 3'UTR carrying point mutations (shown) predicted to disrupt interaction with HSUR2. Cells were transiently transfected with control plasmid (ΔHSUR2), plasmid expressing wild-type HSUR2, or a mutant version of HSUR2 with point mutations that restore binding to the mutant CD69 3'UTR. (**B–D**) Same as in (**A**) with the TP53RK (**B**), STK4 (**C**), and MGA (**D**) luciferase reporters. Dots represent mean values of independent experiments with error bars representing s.d. Two-sided one-sample Student's t-test vs. control (ΔHSUR2) set at 1.0. Sample size (n) is indicated below the chart in each case. *p<0.05, ***p<0.001.

DOI: https://doi.org/10.7554/eLife.50530.012

*Figure 4 continued on next page*

*Figure 4 continued*

The following source data and figure supplements are available for figure 4:

**Source data 1.** Source data for *Figure 4*.
DOI: https://doi.org/10.7554/eLife.50530.015
**Figure supplement 1.** iRICC identifies functional HSUR2 binding sites.
DOI: https://doi.org/10.7554/eLife.50530.013
**Figure supplement 1—source data 1.** Source data for *Figure 4—figure supplement 1*.
DOI: https://doi.org/10.7554/eLife.50530.014

can no longer bind to miR-16 and should therefore repress that target mRNA in a miR-16-independent manner. As expected, inhibition of miR-142–3p activity with locked nucleic acid (LNA) inhibitors ablated HSUR2's ability to repress the expression of all luciferase reporter transcripts tested (*Figure 6A,B* and *Figure 6—figure supplement 2*) (*Gorbea et al., 2017*). Similarly, inhibition of miR-16 ablated HSUR2-mediated repression of reporter transcripts carrying partial PACS1, STK4, MGA, and CD69 3'UTRs (*Figure 6A* and *Figure 6—figure supplement 2A–C*), all of which are bound by HSUR2 without engaging the miR-16BS. In contrast, inhibition of miR-16 with an LNA inhibitor did not affect HSUR2's ability to repress luciferase reporter transcripts carrying partial TP53RK and YTHDC1 3'UTRs (*Figure 6B* and *Figure 6—figure supplement 2D*), both of which contain HSUR2 binding sites that base-pair with the miR-16BS. These results suggest that the arrangement of base pairs between HSUR2 and its target mRNA determines the availability of the miR-16BS and therefore dictates whether HSUR2 utilizes this miRNA for mRNA repression (*Figure 7*).

To test if base-pairing between HSUR2 and its target mRNA determines whether HSUR2 utilizes miR-16 for mRNA repression, we used the PACS1 and TP53RK 3'UTR reporter constructs carrying point mutations in HSUR2 binding sites that result in interactions through different base-pairing arrangements, while maintaining similar binding stabilities (*Figure 5*). Inhibition of miR-16 activity prevented HSUR2 from repressing the wild-type PACS1 reporter transcript (*Figure 6A*) presumably because the miR-16BS is not used to bind PACS1 mRNA, whereas miR-16 no longer affected HSUR2's ability to repress the mutant altbsPACS1mut reporter transcript which now interacted with HSUR2 through its miR-16BS (*Figure 6C*). Conversely, miR-16 activity was not required for HSUR2-mediated repression of the wild-type TP53RK reporter transcript which uses the miR-16BS in its interaction with HSUR2 (*Figure 6B*), but miR-16 activity was required for repression of the altbsTP53RKmut reporter transcript, which was redesigned to interact with HSUR2 without base-pairing with its miR-16BS (*Figure 6D*). Thus, we were able to induce or relieve miR-16 dependence by altering the availability of the miR-16BS in HSUR2. These results indicate that base-pairing between HSUR2 and target mRNAs dictates the availability of the miR-16BS and therefore the requirement of miR-16 activity for HSUR2-mediated mRNA repression.

## Discussion

### iRICC identifies RNA-RNA interactions in vivo

RNA-RNA interactions govern critical processes in all gene expression programs. However, reliably identifying new, biologically relevant interactions between different RNA species has historically posed a challenging problem. Several methods based on psoralen-mediated crosslinking and proximity ligation have been developed recently to identify RNA-RNA interactions (*Aw et al., 2016*; *Lu et al., 2016*; *Sharma et al., 2016*). All these methods rely on proximity ligation of psoralen-crosslinked RNAs, high-throughput sequencing, and identification of chimeric reads to infer RNA-RNA interactions. Although all these methods have successfully identified well-characterized inter-molecular RNA interactions, it is uncertain how reliable they are to detect new, uncharacterized, biologically relevant RNA-RNA interactions. The requirement for reversal of the psoralen crosslinks before library preparation makes these methods unsuitable for determining the nucleotides involved in psoralen-mediated crosslinking and thus cannot distinguish bona fide, in vivo RNA-RNA interactions from RNA duplexes that are generated in vitro. To address this issue, we developed iRICC to identify the RNA binding partners of a single RNA of interest, and the regions and sequences mediating their interactions in vivo (*Figure 1A*). This method relies on psoralen crosslinking-dependent reverse-

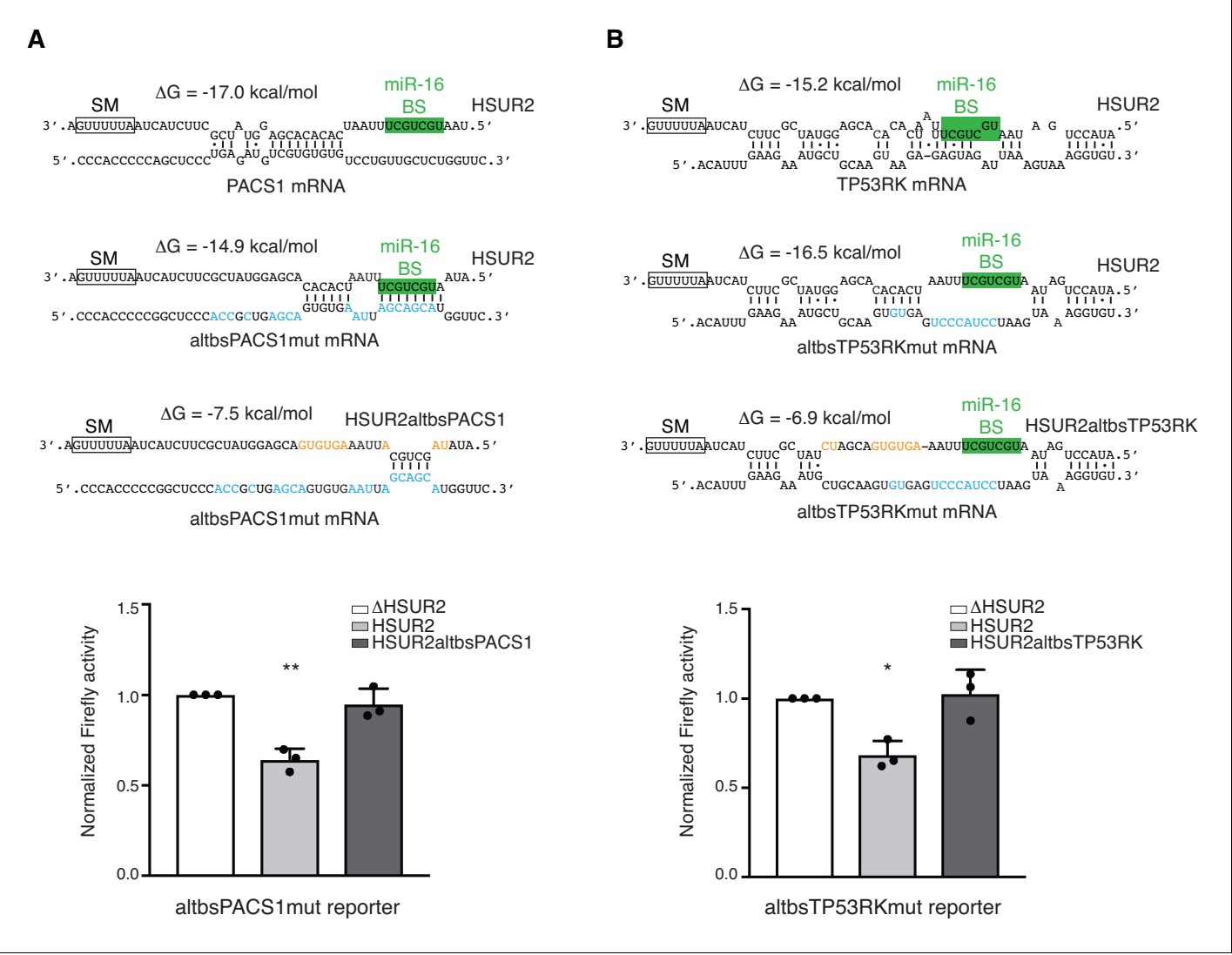

**Figure 5.** HSUR2 binds target mRNAs through a flexible arrangement of base-pairs. (**A**) Same as in *Figure 3B*, but with a mutant version of partial PACS 3′UTR with mutations (blue residues) that result in alternative base-pairing with wild-type HSUR2. Predicted base-pairing between wild-type HSUR2 and wild-type PACS1 mRNA, wild-type HSUR2 and mutant PACS1 mRNA (altbsPACS1mut), and mutant PACS1 mRNA and mutant HSUR2 (HSUR2altbsPACS1) with mutations (orange residues) that disrupt interaction with altbsPACS1mut shown in orange. (**B**) Same as in (**A**), but with partial TP53RK 3′UTR. Dots represent mean values of independent experiments (*n* = 3) with error bars representing s.d. Two-sided one-sample Student's *t*-test vs. control (ΔHSUR2) set at 1.0. *p<0.05, **p<0.01.

DOI: https://doi.org/10.7554/eLife.50530.016

The following source data is available for figure 5:

**Source data 1.** Source data for *Figure 5*.

DOI: https://doi.org/10.7554/eLife.50530.017

transcriptase stalling during library preparation to identify sites of crosslinking and to determine the precise sequences involved in base-pairing.

We decided to focus on interactions between HSUR2 and target mRNAs. For that reason, we performed a selection of polyA+ RNAs as an initial step in iRICC. This step not only constrains the class of target RNAs identified, but also serves as a means to remove ribosomal RNA (rRNA) from samples. A caveat of this version of iRICC is that it most likely misses interactions between HSUR2 and non-polyA+ target RNAs. Alternatively, a rRNA removal step could be used instead of polyA+

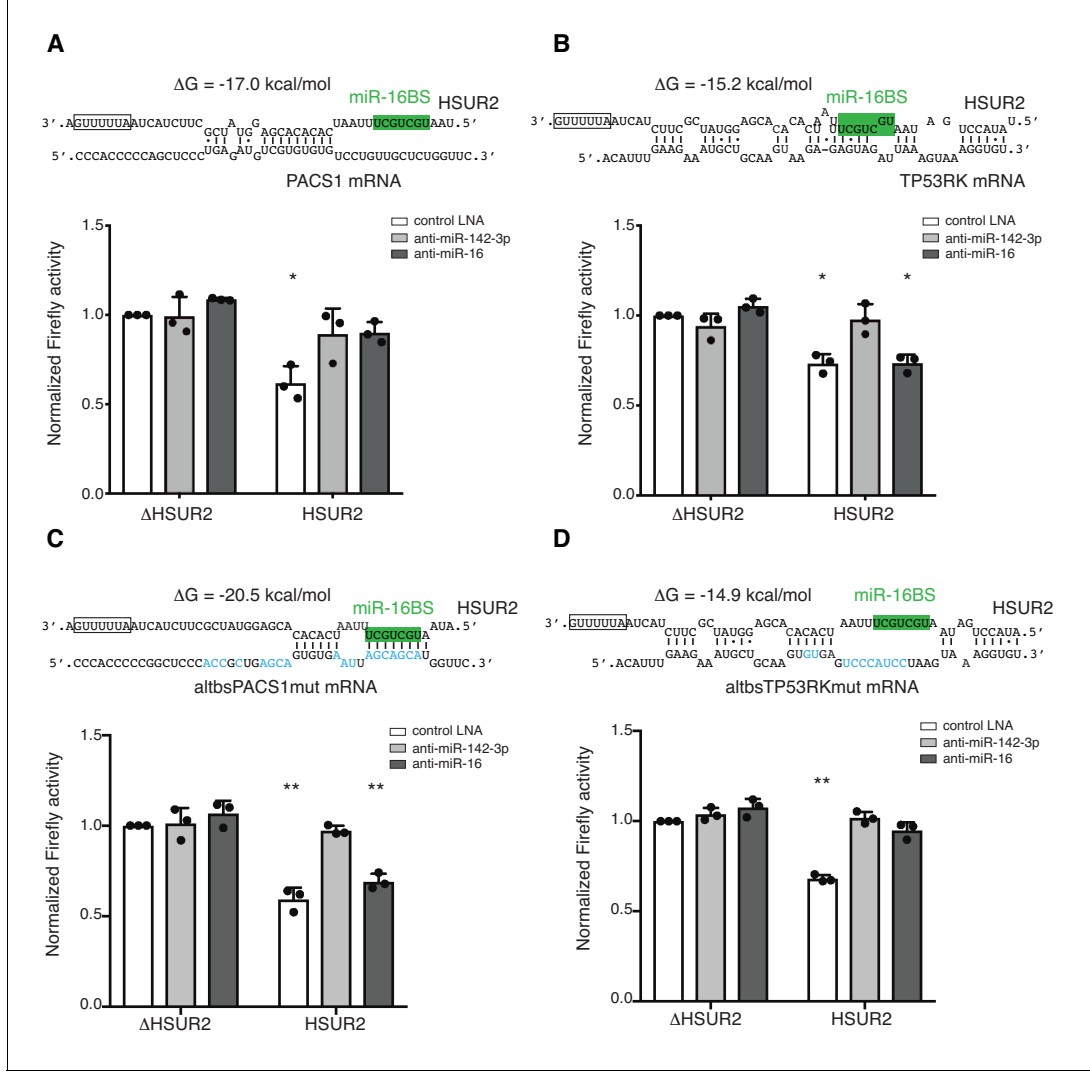

**Figure 6.** Base-pairing with target mRNA dictates the use of miR-16 in HSUR2-mediated mRNA repression. (**A**) U937 cells stably expressing the Firefly luciferase gene with partial PACS1 3'UTR (*Figure 2a*) were transiently co-transfected with a plasmid carrying GFP and either HSUR2 promoter alone (ΔHSUR2) or wild-type HSUR2 (HSUR2) and a control LNA inhibitor, or an LNA inhibitor with complementarity to miR-142–3p or miR-16. (**B–D**) Same as in (**A**) but with cells stably expressing the Firefly luciferase gene with partial TP53RK 3'UTR (**B**), mutant PACS1 (altbsPACS1mut) 3'UTR (**C**), or mutant TP53RK (altbsTP53RKmut) 3'UTR (**D**). Modified residues are shown in blue. Dots represent mean values of independent experiments (*n* = 3) with error bars representing s.d. Two-sided one-sample Student's t-test vs. control (ΔHSUR2) set at 1.0. *p<0.05, **p<0.01.
DOI: https://doi.org/10.7554/eLife.50530.018

The following source data and figure supplements are available for figure 6:

**Source data 1.** Source data for *Figure 6*.
DOI: https://doi.org/10.7554/eLife.50530.022

**Figure supplement 1.** Expression of mutant versions of HSUR2 used in this study.
DOI: https://doi.org/10.7554/eLife.50530.021

**Figure supplement 2.** Base-pairing with target mRNA determines the use of miR-16 in HSUR2-mediated mRNA repression.
DOI: https://doi.org/10.7554/eLife.50530.019

**Figure supplement 2—source data 1.** Source data for *Figure 6—figure supplement 2*.
DOI: https://doi.org/10.7554/eLife.50530.020

selection as a less biased approach for the identification of RNA-RNA interactions for other classes of RNAs.

iRICC defined the sequences mediating interactions between HSUR2 and 171 binding sites in target mRNAs (*Figure 2* and *Supplementary file 2*). Since psoralen derivatives preferably crosslink

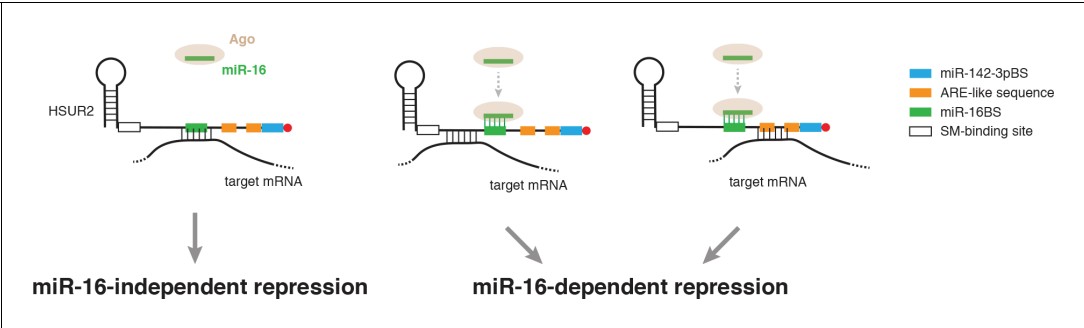

**Figure 7.** Model for alternative use of miR-16 in HSUR2-mediated mRNA repression. HSUR2 base-pairs with target mRNAs through different regions and/or arrangements of base pairs in each case. Engagement of the miR-16 binding site in HSUR2 in the interaction with the target mRNA prevents binding of miR-16 to HSUR2, resulting in miR-16-independent target mRNA repression. When the miR-16 binding site in HSUR2 is not engaged in the interaction with the target mRNA, miR-16 binds to HSUR2 which now represses the mRNA in a miR-16-dependent manner.

DOI: https://doi.org/10.7554/eLife.50530.023

adjacent opposite pyrimidine bases in double-stranded regions (*Bachellerie and Hearst, 1982*; *Cimino et al., 1985*; *Thompson et al., 1982*), we predict that interactions mediated by GC-rich regions will be under-represented in datasets generated by iRICC. Alternatively, different RNA crosslinking reagents (*Harris and Christian, 2009*) could be used in the iRICC protocol in the future to address this bias. Mutational analyses showed that HSUR2 exerts its repressive function through the sequences identified by iRICC in every case tested (*Figure 4* and *Figure 4—figure supplement 1*). iRICC therefore successfully identified functional RNA-RNA interactions in vivo, providing a reliable and general solution to the long-standing problem of identifying unknown, functional RNA-RNA interactions. Although iRICC can confidently determine base-pairing in the proximity of cross-linked residues, confidence in the predicted base-pairing diminishes as the distance between the predicted base pairs and the crosslinked residues increases. Thorough mutational analyses are thus required to determine the full extent of base-pairing for each interaction identified by iRICC.

Gene set enrichment analyses of targets identified by iRICC indicated that HSUR2 regulates the expression of genes with roles in processes relevant to viral infection (*Figure 2—figure supplement 1D* and *Supplementary file 4*) as previously described (*Gorbea et al., 2017*). However, only three mRNAs previously described by RICC-seq (*Gorbea et al., 2017*) as targets of HSUR2 were identified by iRICC, suggesting that both techniques underestimate the number of mRNAs targeted by HSUR2. Substantial differences in protocol design, high-throughput library preparation systems, and criteria utilized for determining target mRNAs presumably account for the limited overlap between the target mRNAs generated by these two distinct protocols.

## Target recognition by HSUR2

HSUR2's primary function is to deliver miRNAs to mRNAs (*Gorbea et al., 2017*). It is therefore not surprising that HSUR2 binding sites are found primarily in 3'UTRs (*Figure 2B*) where most miRNA binding sites occur (*Chi et al., 2009*; *Hafner et al., 2010*). It is difficult to discern the functions of HSUR2 binding sites in coding sequences as well as in 5'UTRs of target mRNAs (*Figure 2B*). It is possible that HSUR2-mediated recruitment of miRNAs to coding sequences contribute to mRNA repression since miRNAs can downregulate mRNA expression by binding to such regions (*Brümmer and Hausser, 2014*; *Fang and Rajewsky, 2011*; *Schnall-Levin et al., 2010*); it is also conceivable that HSUR2 regulates target mRNAs through miRNA-independent mechanisms when binding to coding sequences and 5'UTRs. Further experimentation will be required to explore these possibilities.

The mechanism by which HSUR2 can specifically recognize binding sites in target mRNAs remains unknown. The fact that HSUR2 does not present a specialized or seed region that base-pairs with most binding sites (see below) suggest that sequence complementarity is required but not sufficient for the selection of target mRNAs. This characteristic is shared by cellular Sm-class RNAs. For example, base-pairing between U1 snRNA and the 5' splice site and between U2 snRNA and the branch-point sequence are essential for splicing (*Black et al., 1985*; *Mount et al., 1983*; *Parker et al., 1987*; *Séraphin et al., 1988*; *Siliciano and Guthrie, 1988*; *Wu and Manley, 1989*; *Zhuang and*

*Weiner, 1986*; *Zhuang and Weiner, 1989*). Yet, *cis*-acting signals like splicing enhancers, and *trans*-acting factors like SR proteins are essential for guiding and escorting spliceosomal small nuclear ribonucleoprotein particles (snRNPs) to the appropriate splicing signals before base-pairing is established (*Kohtz et al., 1994*; *Lavigueur et al., 1993*; *Roscigno and Garcia-Blanco, 1995*; *Sun et al., 1993*; *Tarn and Steitz, 1995*; *Tian and Maniatis, 1993*). This combination of protein-assisted recruitment of snRNPs and RNA-RNA base-pairing results in exquisite specificity as well as a enormous regulatory potential, which is manifested by the immense protein diversity that is generated by alternative splicing (*Nilsen and Graveley, 2010*). It is conceivable that, similarly to cellular snRNPs, additional *cis*- and *trans*-acting factors may assist HSUR2 in the recognition of binding sites in target mRNAs increasing both the specificity and versatility of HSUR2-mediated mRNA regulation. Further research will be required to test this hypothesis.

## HSUR2 does not present a seed region

Unlike miRNAs and other regulatory small ncRNAs that function as guides that base-pair with target RNAs to deliver effector complexes (*Gorski et al., 2017*; *Künne et al., 2014*), HSUR2 does not use a dedicated, or seed region to interact with target mRNAs. iRICC revealed that HSUR2 can use different regions of its sequence to interact with different targets and can also employ diverse arrangements of base pairs to mediate these interactions (*Figure 1C,D* and *Figure 1—figure supplement 2*; *Supplementary file 2*). This property is shared with cellular Sm-class RNAs, which allow for variability in both the choice of region engaged in base-pairing as well as in the arrangement of base pairs between one region of the Sm-class RNA and different target RNAs (*Cotten et al., 1988*; *Roca and Krainer, 2009*; *Roca et al., 2013*; *Staley and Guthrie, 1998*). Mutational analyses showed that HSUR2 can interact with the same target through different arrangement of base pairs (*Figure 5*), pointing to the stability of the interaction as the main variable determining the outcome of HSUR2-mediated mRNA regulation. In the future, it will be interesting to determine if base-pairing strength explicitly determines the extent of HSUR2-mediated mRNA repression.

We hypothesize that the binding properties of HSUR2 confer this viral ncRNA with potential advantages over seed-based systems. For small regulatory ncRNAs that employ seed-based mechanisms for target regulation, the repertoire of regulated transcripts is constrained by the presence of a discrete sequence complementary to the seed region of the ncRNA, in the same way that the size of a wrench constrains the size of bolts and nuts that can be adjusted with it. Several wrenches of different sizes are required to adjust the diverse collection of nuts and bolts that are part of an engine. By analogy, most herpesviruses express large numbers of miRNAs to regulate a diverse collection of transcripts during latency to generate a cellular environment that is conducive to viral persistence (*Skalsky and Cullen, 2010*). We propose that flexible base-pairing allows HSUR2 to regulate a diverse group of transcripts functioning like a 'crescent wrench' that can be adjusted to fit nuts and bolts of different sizes. By using ncRNAs that employ 'non-seed' or flexible base-pairing, HVS can potentially minimize the number of regulatory ncRNAs required to effectively regulate a defined set of transcripts and therefore minimize the investment in viral genome space.

Seed-based RNA regulatory systems are sensitive to mismatches in interactions between the seed region of the ncRNA and target transcripts (*Bartel, 2018*; *Chipman and Pasquinelli, 2019*; *Gorski et al., 2017*; *Künne et al., 2014*). Single mismatches in the seed region of miRNAs can dramatically reduce their association with target transcripts (*Chandradoss et al., 2015*; *Salomon et al., 2015*; *Schirle et al., 2014*). Thus, single point mutations in binding sites can render regulation by miRNAs ineffectual, making it relatively easy for the host to escape viral miRNA-based regulation. We have shown that extensive mutagenesis of HSUR2 binding sites does not affect HSUR2 binding as long as new base pairs are formed to maintain a similar overall binding affinity (*Figure 5*). We speculate that this binding flexibility allows HSUR2-mediated regulation to withstand considerable target sequence evolution, making it difficult for the host to escape this viral mechanism of gene regulation.

## HSUR2 differentially recruits miR-16 to a subset of target mRNAs

The molecular mechanism through which miR-142–3p contributes to repression of all HSUR2 target mRNAs tested so far is currently unknown. Since this miRNA is abundantly expressed in hematopoietic tissues and mostly absent in non-hematopoietic ones (*Chen et al., 2004*), it is conceivable that

HSUR2 would not be able to repress target mRNAs in non-hematopoietic tissues, curbing the ability of HVS to establish latent infections in non-hematopoietic tissues and thus defining the tropism of this virus' latent infection.

Flexible base-pairing between HSUR2 and target mRNAs provides a mechanism for the differential recruitment of miR-16 to a select group of transcripts (*Figure 6*, *Figure 7*, and *Figure 6—figure supplement 2*). GO-term enrichment analysis showed that miR-16-dependent HSUR2 targets encode proteins with roles in apoptosis, consistent with the fact that the interaction between miR-16 and HSUR2 is required for inhibition of apoptosis by this viral Sm-class RNA (*Gorbea et al., 2017*). The miR-16 family regulates the cell cycle by modulating the expression of genes with critical roles in the G1-S transition including cyclins D1, D2, D3, E1, and CDK6 (*Bonci et al., 2008*; *Cimmino et al., 2005*; *Liu et al., 2008*). Interestingly, the abundance of members of this family of miRNAs is dynamically regulated by the cell cycle, with the highest abundance of these miRNAs observed in cells that are arrested in G0 (*Rissland et al., 2011*). Assuming that the abundance of these miRNAs determines the extent to which HSUR2 can affect its targets, we speculate that this mechanism allows HSUR2 to efficiently repress a subset of transcripts in a cell cycle-dependent manner. This provides a mechanism for HVS to regulate the expression of key genes in specific situations, like when the latently infected cell tries to arrest in G0. Further investigation is required to explore these exciting possibilities.

## Materials and methods

### Plasmids

For stable expression of the Firefly luciferase gene fused to partial 3′UTRs of HSUR2 target mRNAs containing single HSUR2 binding sites, the thymidine kinase promoter from pGL4.54[luc2/TK] (Promega), the optimized *luc2* gene from pmirGLO (Promega), and the bovine growth hormone polyadenylation signal (bGH polyA) from PCDNA3 (Invitrogen) were cloned into pLenti CMVTRE3G eGFP Puro (Addgene #27570) to generate the pLenti-TK-Firefly-Control plasmid. Fragments corresponding to full-length BCCIP 3′UTR or partial TP53RK, CD69, MGA, STK4, PACS1, and YTHDC1 3′UTRs were amplified by PCR from marmoset genomic DNA and cloned between the *luc2* gene and the bGH polyA in pLenti-TK-Firefly-Control. For stable expression of *Renilla* luciferase, the *Renilla* luciferase gene from pmirGLO was cloned into LeGO-G/BSD (Addgene #27354). All these plasmids were deposited in Addgene. Mutant versions of these plasmids were generated by site-directed mutagenesis, and confirmed by sequencing. Plasmids expressing GFP plus wild-type HSUR2 (pBS-GFP-HSUR2) or GFP plus the HSUR2 promoter alone (pBS-GFP-ΔHSUR2) have already been described (*Gorbea et al., 2017*). Mutant versions of HSUR2 were obtained by site-directed mutagenesis of the plasmid pBS-GFP-HSUR2, and confirmed by sequencing.

### Cell culture

Common marmoset cj319-WT (*Cook et al., 2004*) and cj137-WT (*Gorbea et al., 2017*) T cells infected with wild type *Herpesvirus saimiri* were grown in RPMI 1640 medium (ThermoFisher Scientific) supplemented with 20% fetal bovine serum (FBS), 100 U/ml of penicillin, 100 μg/ml of streptomycin, 1 ml of normocin (InvivoGen), 2 mM glutamax (ThermoFisher Scientific), 1 mM sodium pyruvate (ThermoFisher Scientific) and 0.1% antioxidant supplement (Sigma Aldrich, cat# A1345). 293T/17 cells were obtained from ATCC (CRL 11268) and grown in Dulbecco's-modified Eagle's medium (DMEM) supplemented with 10% FBS, 100 U/ml of penicillin, 100 μg/ml of streptomycin and 1 mM sodium pyruvate. U-937 histiocytic lymphoma cells were also obtained from ATCC (CRL-1593.2) and grown in RPMI 1640 medium supplemented with 10% FBS, antibiotics, glutamax and sodium pyruvate as described above. Cells lines were not authenticated. All cell lines were routinely tested for the presence of mycoplasma. All lentiviral transductions were performed in U937 cells with supernatants generated by co-transfection of 293T/17 cells with plasmids pMD2.G (Addgene #12259), pMDLG/pRRE (Addgene #12251), pRSV-Rev (Addgene #12253), and a lentiviral targeting vector (described above) harboring cDNA for the gene of interest. Stable cell lines were generated by selection with blasticidin (for *Renilla* luciferase-expressing cells) and puromycin (for *Renilla* and Firefly luciferase-expressing cells). Stable Firefly- and Renilla-luciferase-expressing U937 cells (see

below) were grown in complete medium plus 4 µg/ml of puromycin (InvivoGen) and 10 µg/ml of blasticidin (InvivoGen).

## iRICC

### In vivo RNA-RNA crosslinking

$4 \times 10^8$ cj319-WT and cj137-WT marmoset T cells were washed with phosphate-buffered saline (PBS) and suspended in 4 ml of PBS containing 200 µg/ml 4'-aminomethyl-4,5',8-trimethylpsoralen (AMT, Cayman Chemical). Cells were irradiated at 365 nm on ice for 1 hr, collected and fractionated in 100 µl samples. One hundred microliters of 6 M guanidinium hydrochloride was added to each sample followed by 20 µl of a 20 mg/ml solution of RNAse-free proteinase K (Ambion) and 10 µl of 10% sodium dodecyl sulfate (SDS). Samples were incubated at 65°C for 1 hr. One milliliter of TRIzol (Ambion) was added to each sample and then stored at −70°C until used.

### PolyA+ RNA purification

Total, AMT-crosslinked RNA was purified from samples in TRIzol according to the manufacturer's protocol (Ambion). PolyA+ RNA was isolated using oligo d(T)25 magnetic beads (New England Biolabs) from samples containing 0.98 mg (Experiment #1), 0.93 mg (Experiment #2), and 0.8 mg (Experiment #3) of total RNA. Briefly, purified total RNA was suspended in 600 µl of water and heated to 85°C for 3 min. One volume of 2X binding buffer (40 mM Tris-HCl, pH 7.5, 1 M NaCl, 1% SDS, 2 mM EDTA, 10 mM DTT) at 65°C was added and mixed immediately with 1 ml of oligo d(T)25 magnetic beads washed twice with 1 ml of 1X binding buffer. Samples were incubated with continuous rotation for 30 min at room temperature. Beads were washed twice for 1 min with 1 ml of wash buffer 1 (20 mM Tris-HCl, pH 7.5, 500 mM NaCl, 0.1% SDS, 1 mM EDTA, 5 mM DTT) and twice with 1 ml of wash buffer 2 (20 mM Tris-HCl, pH 7.5, 500 mM NaCl, 1 mM EDTA) at room temperature. Beads were then incubated with 1 ml of low-salt buffer (20 mM Tris-HCl, pH 7.5, 200 mM NaCl, 1 mM EDTA) for 1 min at room temperature. PolyA+ RNA was eluted from the beads with 300 µl of elution buffer (20 mM Tris-HCl, pH 7.5, 1 mM EDTA) at 50°C for 5 min with continuous agitation. PolyA+ RNA was precipitated by adding 5 µl of a 10 mg/ml solution of glycogen, 30 µl of 3 M sodium acetate pH 5.2, and 1 ml of 100% ethanol, and incubated overnight at −20°C. PolyA+ RNA was recovered by centrifugation at 17,000 x *g* for 30 min at 4°C, washed with 80% ethanol, centrifuged at 17,000 x *g* for 15 min at 4°C, and suspended in 100 µl of water. The amounts of purified polyA+ RNA recovered were 23.2 µg (Experiment #1), 23.3 µg (Experiment #2) and 15 µg (Experiment #3).

### mRNA fragmentation

Twelve microliters of 10X annealing buffer (200 mM Tris-HCl, pH 7.5, 300 mM NaCl), 4 µl of a 20 pmoles/µl solution of DNA oligonucleotide with perfect complementarity to the entire HSUR2 sequence (anti-H2 DNA), and 4 µl of water were added to 100 µl of purified polyA+ RNA. Each sample was then aliquoted into four separate PCR tubes (30 µl per tube). Annealing was performed in a Bio-Rad T100 thermal cycler by incubating tubes at 95°C for 5 min, temperature was then decreased to 65°C with a ramp rate of 0.1 °C s$^{-1}$ and held at 65°C for 5 min. Next, temperature was decreased to 25°C with a ramp rate of 0.1 °C s$^{-1}$, held at 25°C for 5 min, and then decreased further to 4°C with a ramp rate of 0.1 °C s$^{-1}$, and held indefinitely at 4°C. Eight microliters of 5X S1 buffer (ThermoFisher Scientific) and two units (2 µl) of S1 nuclease (ThermoFisher Scientific) were added to each tube and incubated for 30 min at room temperature. Sixty microliters of water were then added to each tube, and the contents of each tube were transferred to a 1.5 ml eppendorf tube, followed by addition of 1 ml of TRIzol to inactivate the S1 nuclease. RNA was purified from the TRIzol aqueous phases on RNA clean and concentrator−5 columns following the manufacturer's protocol (Zymo Research) and eluted with 20 µl of water. Sixty-five microliters of water, 10 µl of 10X DNase I buffer (New England Biolabs), and 5 µl (10 units) of DNase I (New England Biolabs) were then added to each tube. Samples were incubated for one hour at 37°C. One milliliter of TRIzol was added to each tube and RNA was purified on RNA clean and concentrator−5 columns from the TRIzol aqueous phases. RNA was eluted from each column in 20 µl of water and pooled (final volume: 80 µl/sample). RNA was then quantified in a NanoDrop spectrophotometer: Experiment #1 yielded 12 µg of RNA, Experiment #2 yielded 14.2 µg of RNA, and Experiment #3 yielded 8.3 µg of RNA.

## RNAse H digestion

Fragmented RNA samples were divided in two (HSUR2 and Control samples), and each sample was then aliquoted into two separate PCR tubes (20 µl per tube). Three microliters of 10X annealing buffer and 7 µl of water were added to each tube corresponding to the HSUR2 sample. Tubes corresponding to the Control samples received 1 µl of anti-H2 DNA (20 pmoles), 3 µl of 10X annealing buffer, and 6 µl of water. Annealing was performed as described above for all tubes. After annealing 14 µl of water, 5 µl of 10X RNase H buffer (New England Biolabs) and 1 µl of RNase H (New England Biolabs) were added to each tube. Tubes were incubated in a Bio-Rad T100 thermal cycler according to the following program: 37°C for 1 hr, 65°C for 20 min, and 4°C indefinitely. After RNAse H digestion, each tube received 35 µl of water, 10 µl of 10X DNase I buffer and 5 µl of DNase I (New England Biolabs) and was incubated at 37°C for 30 min for complete removal of anti-H2 DNA. Samples were then transferred to 1.5 ml eppendorf tubes and 1 ml of TRIzol was added to each sample and stored at −70°C.

## Capture of HSUR2

After RNase H digestion, RNA was recovered from the TRIzol aqueous phase with RNA clean and concentrator−5 columns following the manufacturer's protocol (Zymo Research). RNA was eluted from each column with 20 µl of water and pooled. The amounts of RNA recovered were as follows:

Experiment #1: 4.6 µg (HSUR2 sample) and 4.6 µg (Control sample).
Experiment #2: 5.3 µg (HSUR2 sample) and 4.9 µg (Control sample).
Experiment #3: 3.1 µg (HSUR2 sample) and 3.2 µg (Control sample).

A biotinylated RNA oligonucleotide antisense to HSUR2 (b-antiH2rASO, obtained from Integrated DNA Technologies, at 25 pmol/µl) was denatured at 85°C for 3 min and placed on ice. Fifty picomoles of b-antiH2rASO (*Supplementary file 3*) were added to each sample, and the mixture was incubated at 85°C for 3 min followed by addition of 0.3 ml of LiCl hybridization buffer (10 mM Tris-HCl, pH 7.5, 1 mM EDTA, 500 mM LiCl, 1% Triton X-100, 0.2% SDS, 0.1% deoxycholate, 4 M urea) heated to 85°C. Samples were then incubated at 25°C for 1 hr with continuous agitation at 1200 rpm. After annealing of b-antiH2rASO, samples were transferred to tubes containing 200 µl of magnetic streptavidin C1 dynabeads (Invitrogen) previously washed with 1 ml of LiCl hybridization buffer. Capture of b-antiH2rASO was performed by incubation at 37°C for 1 hr with continuous agitation at 1200 rpm. Tubes were placed on magnetic rack, supernatants were discarded and beads were washed three times at 50°C for 5 min each with low stringency wash buffer [1X SSPE (10 mM $NaH_2PO_4$, pH 7.4, 150 mM NaCl, 1 mM EDTA), 0.1% SDS, 1% NP-40, 4 M urea] and three times at 50°C for 5 min each with high stringency wash buffer [0.1X SSPE (1 mM $NaH_2PO_4$, pH 7.4, 15 mM NaCl, 0.1 mM EDTA), 0.1% SDS, 1% NP-40 and 4 M urea] with continuous agitation at 1200 rpm. RNA was eluted from the beads with 1 ml of TRIzol.

## Recapture of released b-antiH2rASO oligonucleotide

RNA was recovered from the TRIzol aqueous phases with RNA clean and concentrator−5 columns and eluted with 20 µl of water. To release b-antiH2rASO from the purified RNA, samples were heated to 85°C for 5 min and immediately diluted with 180 µl of TE buffer, pH 7.5 heated to 85°C. Samples were transferred immediately to tubes containing 200 µl of magnetic streptavidin C1 beads previously washed with TE, pH 7.5 and incubated at 42°C for 30 min with continuous agitation at 1200 rpm. Tubes were placed on magnetic rack and the supernatant fractions were collected and purified on RNA clean and concentrator−5 columns using two volumes (400 µl) of RNA binding buffer (Zymo Research) plus one volume (600 µl) of 100% ethanol. RNA was eluted from the columns with 11 µl of water (recovered ~9.5 µl of each sample) and 1 µl of each sample was quantified using the Qubit RNA HS Assay kit (Invitrogen). Since the amount of RNA in each sample was below the limit of detection, 1 µl of each sample was mixed with 19 µl of control RNA (Invitrogen) diluted 1 to 20 (0.5 ng/µl) with Qubit RNA HS buffer. The amount of RNA in each sample was then estimated from the difference in the amount of RNA in each spiked sample and the amount of RNA in a reference sample prepared with 19 µl of diluted control RNA plus 1 µl of water. The amount of purified RNA in each sample used for high-throughput sequencing library preparation was:

Experiment #1: 4.1 ng (HSUR2 sample) and 3 ng (Control sample).
Experiment #2: 3.6 ng (HSUR2 sample) and 3.7 ng (Control sample).

Experiment #3: 4.5 ng (HSUR2 sample) and 5.4 ng (Control sample).

## Library preparation and high-throughput sequencing

Samples for high-throughput sequencing for all three iRICC experiments were prepared in parallel from 8 µl of RNA for each sample using the SMARTer Stranded Total RNA-seq Kit v2 – Pico Input Mammalian (Takara Bio USA) following the user manual, but with a few modifications. Briefly, option 2 ("Starting from highly degraded RNA) of Part A of the protocol was used. In part C of the protocol ('Purification of the RNA-Seq Library Using AMPure Beads'), DNA was eluted in 20 µl of water and proceeded directly to Part E ("PCR2-Final RNA-Seq Library Amplification) where libraries were amplified for 14 cycles, omitting Part D of the protocol. After final PCR amplification, libraries were purified with DNA Clean and Concentrator −5 (Zymo Research) following the manufacturer's instructions and eluted in 12 µl of the Elution Buffer provided. HiSeq 125 Cycle Paired-End sequencing was performed by the High Throughput Genomics Core Facility at the Huntsman Cancer Institute (University of Utah) on an Illumina HiSeq 2500 instrument.

## Bioinformatic analyses for HSUR2 target enrichment

A reference sequence file was created by combining the genomic sequences of *Callithrix jacchus* (v3.2.1) and Saimiriine herpesvirus 2 (Genbank ID: X64346.1). A gene annotation file was created by combining *Callithrix Jacchus* Ensembl annotations (v91) and gene definitions from X64346.1. The STAR (v2.5.3a) 'genomeGenerate' command was used to create an index file using the reference and annotation files described above and with 'sjdbOverhang' set to 124. Reads were processed using cutadapt (v1.6) to remove the first three base pairs at the start of the read and the ends of reads beginning at polyA stretches of 10 bp or longer. Reads shorter than 10 bp after processing were removed from the analysis. Trimmed reads were aligned to the reference index using STAR and set to report alignment data in bam format. The Subread 'featureCount' command was used to generate read counts for each gene in the annotation file described above, only counting reads on the same strand as the annotation. DESeq2 (v1.14.1) was used to normalize gene counts across the samples and determine differential expression between the HSUR2 and control samples in each replicate.

## Identification of HSUR2 target mRNAs

A list of HSUR2 targets was generated by filtering the list of genes sequenced according to the following criteria: a) genes with fewer than eight reads across all replicates were removed; b) genes with a non-adjusted p value > 0.05 were eliminated; and c) only genes with an enrichment of 1.5-fold and higher (log2 fold change ≥0.6) in the HSUR2-positive pulldown *versus* the Control (HSUR2-negative) pulldown were used in further analyses. In an initial list of 220 potential HSUR2 targets that conformed to the above criteria, six targets corresponded to snoRNAs, one mRNA encoded a *Herpesvirus saimiri* protein and one was a pseudogene. These were not considered for further analysis. Thus, we initially identified 212 host mRNAs of which 120 displayed peaks of aligned reads with clear boundaries and were subjected to the analyses described below.

## Determination of potential crosslinking sites and sequences mediating the interaction between target mRNAs and HSUR2

The coordinates of genes included in the list of HSUR2 targets meeting the criteria described above and the assembled marmoset genome were inserted into the IGV_2.4.2 genome browser to visually inspect the peaks of aligned reads (using both read1 and read2) across each gene and across all experiments. From the initial list of 212 potential HSUR2 target mRNAs a total of eight (3.6%) had unannotated 3'UTRs and were eliminated from further analyses. A total of 32 mRNAs (14.5%) showed high background in control samples and lacked clear and discrete peaks of aligned reads that could be identified in all three experiments and thus were not considered any further. Fifty-two mRNAs (23.6%) had clear peaks of aligned reads across the three experiments, but lacked consistent boundaries (i.e. similar abrupt or smooth edges or drops of reads within a peak in the three experiments). These mRNAs were not further analyzed since crosslinking sites could not be unambiguously determined. The majority of the remaining 120 target mRNAs exhibited peaks with abrupt edges or a sharp drop of aligned reads within a peak, which we used as the primary indicator of a potential

site of crosslinking. In a minority of the cases (~17%), peaks were symmetrical in appearance and lacked abrupt edges. To determine the sequences that might be involved in base-pairing with HSUR2, an initial window of 40 nucleotides centered at the potential site of crosslinking (indicated by a sharp drop of reads, or by smooth edges of symmetrical peaks) was entered into the *RNAcofold* web server from the ViennaRNA Package 2.0 at the Institute of Theoretical Chemistry at the University of Vienna (http://rna.tbi.univie.ac/at/RNAcofold) (*Bernhart et al., 2006*) along with the full-length sequence of HSUR2 using default parameters. The initial results indicated that the sequences in HSUR2 including the Sm protein binding site (AUUUUUG) or forming the adjacent stem-loop structure were not involved in heteroduplex formation with a target mRNA in all the cases analyzed. Next, the window of sequence in the target mRNA was iteratively increased to 60 and 80 (for long binding sites) nucleotides until the results of two analyses (i.e. 40 and 60 nucleotides or 60 and 80 nucleotides) yielded similar binding patterns between the target mRNA sequence and HSUR2. The 'minimal' sequences mediating the interaction of target mRNAs and HSUR2 were then determined in *RNAcofold* after trimming the sequences not predicted to be involved in contiguous (i.e. not separated by internal secondary structures) base-pairing between the two RNA species. The final analysis was also used to obtain an estimated ΔG of binding and minimal free energy for the interaction of the minimal HSUR2-binding site in the target mRNA with the minimal HSUR2 sequence base-pairing with it. The interactions between the HSUR2 binding sites in target mRNAs and corresponding HSUR2 sequences were visualized using the *RNAfold* web server from the ViennaRNA Package 2.0 (http://rna.tbi.univie.ac/at/RNAfold) (*Lorenz et al., 2011*) to determine the most likely site of crosslinking between the two sequences (i.e. according to the sharp drop of aligned reads or the smooth edges of symmetrical peaks). To this end, we inserted five nucleotides (e.g. AAAAA or UUUUU) between the two interacting sequences and folded the sequence onto itself ensuring that the inserted nucleotides formed a connecting loop. This confirmed that potentially crosslinked pyrimidines were present at the end of helical runs or at G-U wobble base-pairs, sites at which psoralen is most reactive (*Bachellerie and Hearst, 1982*; *Thompson et al., 1982*). From the initial list of 120 potential HSUR2 target mRNAs that exhibited peaks of aligned reads with similar abrupt or smooth edges, we could not determine potential base-pairing between local sequence of 10 mRNAs and HSUR2 using *RNAcofold*. A final list of high confidence HSUR2 target genes is shown in *Supplementary file 1*.

## Transfections

Ten million cj319-WT marmoset T cells were nucleofected in a nucleofector 2b device using Amaxa's human T-cell kit (Lonza) and program X-001 with 1 nmol of chimeric ASOs (Integrated DNA Technologies) containing a backbone of phosphorothioate linkages and five nucleotides on each end substituted with 2'-*O*-methoxyethyl ribonucleotides and directed to either GFP (5'-UCACCTTCACCCTC TCCACU-3') (control sample) or to HSUR2 (5' AAGCGATACCTCGTGUGUGA3'), resulting in the specific degradation of HSUR2 (*Cazalla et al., 2010*; *Ideue et al., 2009*). Twenty-one hours after transfection, the cells were harvested, washed with PBS and stored in TRIzol at −80°C until used. Two million U937 cells expressing Firefly and *Renilla* luciferases were nucleofected with Amaxa's kit V (Lonza) and program T-020 using 3 μg of either vector encoding GFP plus wild type HSUR2, GFP plus the HSUR2 promoter alone (ΔHSUR2), or GFP plus a mutant version of HSUR2. Co-transfections of U937 cells with a HSUR2 plasmid and a locked nucleic acid (LNA) microRNA inhibitor were performed with 3 μg of plasmid DNA plus 50 pmoles of either control (mirVana miRNA inhibitor negative control #1 cat# 4464076, Ambion) or anti-miR-142–3p (hsa-miR-142–3p miRCURY LNA inhibitor cat# 4100271–001, Exiqon) or anti-miR-16 (hsa-miR-16-mirVana cat# 4464084, Ambion) LNAs. Transfected U937 cells were grown for 18–21 hr after nucleofection and FACS-sorted on a BD FACSAria 4-laser cell sorter (BD Biosciences) equipped with BD FACSDiva v8.0 software using an 85 μm nozzle and 407 nm (V 450/50 filter set) and 488 nm (B 530/30 filter set) lasers for DAPI-negative and GFP-positive cells. Immediately after sorting cells were incubated for 1 hr at 37°C/5% $CO_2$, centrifuged at 1000 x *g* for 20 min and then used in dual luciferase assays as described below. 293T/17 cells growing on 6-well plates were transfected with 2.5 μg of plasmid expressing either GFP plus the HSUR2 promoter, GFP plus wild type HSUR2, or GFP plus a mutant version of HSUR2 and 2.5 μg of a plasmid expressing GFP plus HSUR7 using 7.5 μl of lipofectamine 3000 reagent and 10 μl of P3000 reagent following the manufacturer's protocol (Invitrogen). Cells were incubated for 18 hr after transfection at 37°C/5% $CO_2$, harvested with 2 ml of TRIzol and stored at −80°C until used.

## Luciferase reporter assays

Thirty thousand GFP-positive DAPI-negative U937 cells were FACS-sorted into 0.5 ml of complete culture medium and recovered by centrifugation as described above. Media was removed by aspiration and the cell pellet was lysed with 50 µl of passive lysis buffer (Promega) for 20 min at room temperature with occasional vortexing. Ten microliters of each sample were then plated in triplicate per well of a 96-well plate and analyzed for Firefly luciferase activity using 50 µl of luciferase II substrate reagent (Promega). Luminescence in each well was measured in a Veritas microplate luminometer (Turner Biosystems) equipped with Glomax v1.9.3 software using an integration time of 5 s. Immediately after, 50 µl of Stop and Glo reagent (Promega) was added to each well and *Renilla* luciferase activity was measured as described above. Normalized light units (NLU) were obtained by dividing the Firefly luciferase relative light units (RLU) by the corresponding *Renilla* RLU of each well. The relative luciferase activity was then calculated by dividing the NLU of each sample by the average NLU of the control sample (+ΔHSUR2), set as 1.0.

## Northern blotting

Total RNA from cj319-WT marmoset cells or 293T/17 cells transfected with wild type or mutant versions of pBS-GFP-HSUR2 plasmids and stored in 1 ml of TRIzol was prepared following the manufacturer's protocol (Ambion). RNA samples were then suspended in 20 µl (marmoset cells) or 50 µl (293T/17 cells) of 2 mM Tris-HCl, pH 7.5, 8 M urea and 20 mM EDTA and heated at 65°C before loading onto a 6% denaturing polyacrylamide gel (only 25 µl of each 293T/17 cell sample were loaded onto the gel). Gels were run at 10 W in 1X TBE and transferred onto a Zeta-Probe nylon membrane (Bio-Rad) for 30 min at 1 A using a semi-dry blotting unit (Fisher Biotech). The membranes were pre-hybridized at 42°C in ExpressHyb hybridization solution (Clontech) for 30 min. Hybridization of $^{32}$P-labeled probes to HSUR2, HSUR7 and U6 was performed in ExpressHyb solution overnight at 42°C. Blots were washed once with 2X saline sodium citrate (SSC) buffer, 0.1% SDS for 15 min at room temperature followed by a wash with 0.5X SSC, 0.1% SDS for 15 min at room temperature. Membranes were wrapped in saran wrap and exposed to a phosphorImager screen (GE Healthcare).

## Probe radiolabeling

Radiolabeled probes were prepared in 20 µl reactions containing 10 pmoles of DNA probes to HSUR2, HSUR7 or U6 (*Supplementary file 3*), 10 units of T4 polynucleotide kinase (New England Biolabs) and 151.5 µCi of [γ-$^{32}$P]ATP (6000 Ci/mmol, PerkinElmer) and incubated at 37°C for 1 hr. Unincorporated isotope was removed by centrifugation using Mini Quick G-25 gel filtration columns following the manufacture's protocol (Roche). Radiolabeled probes were eluted in a volume of 50 µl of water and 20 µl were used in each hybridization experiment.

## Quantitative RT-PCR

Total RNA was purified from cj319-WT cells stored in TRIzol according to the manufacturer's instructions (Ambion). RNA was suspended in 85 µl of water, 10 µl of 10X DNAse reaction buffer (New England Biolabs), and 10 units (5 µl) of RNAse-free DNAse I (New England Biolabs) and incubated at 37°C for 1 hr. cDNA was synthesized in 40 µl reactions from 1.8 µg of DNase I-treated total RNA using the High-Capacity cDNA Reverse Transcription Kit with MultiScribe Reverse Transcriptase and random primers (Applied Biosystems). Subsequently, real-time PCR was performed in 8 µl reactions using primers (see *Supplementary file 3*) at 0.5 µM, cDNA diluted at a ratio of 1:5 (except MGA for which the cDNA was used undiluted, or for β-actin for which the cDNA was diluted 1:5,000, or for ribosomal 18 s for which the cDNA was diluted 1:50,000) and KAPA SYBR green (KAPA Biosystems) in a Roche 480 Light Cycler (95°C for 5 min, one cycle; 95°C for 10 s, 60°C for 10 s and 72°C for 10 s, 55 cycles followed by a melting curve of the reaction product from 65°C to 97°C with a ramp rate of 0.11 °C s$^{-1}$). qPCR primers were designed using Primer3Plus (https://primer3plus.com/cgi-bin/dev/primer3plus.cgi) and tested using cDNA prepared in 20 µl reactions from 900 ng of DNase I-treated total RNA and diluted at ratios 1:1–1:5000. All reactions were performed in triplicate in each independent experiment. Relative expression of HSUR2 target mRNAs was calculated as previously described (*Gorbea et al., 2017*).

## Statistical analyses

No statistical methods were used to predetermine sample size, nor were the experiments randomized or the investigators blinded to sample allocation during experiments and evaluation of experimental results. 'Biological replicates' (*n*), indicated in figures and figure legends, refers to the number of independent experiments performed. The number of independent experiments was chosen to allow for statistical significance. Statistical analysis was performed using Graphpad Prism 7. Two-sided *P* values of biological replicates shown in *Figure 3B*, *Figure 4*, *Figure 5*, *Figure 6*, *Figure 4—figure supplement 1B*, and *Figure 6—figure supplement 2B* were obtained with one sample Student's *t*-tests compared to the control samples set at 1.0. Statistical analysis of biological replicates shown in *Figure 1—figure supplement 1B* was performed with Student's *t*-test corrected for multiple comparisons with the Holm-Sidak method (alpha = 0.05). The pairwise comparison made between HSUR2 targets and controls shown in *Figure 2D* was performed using unpaired, two-tailed Student's *t*-test with Welch's correction for unequal variances, after using the F test to determine whether the variance of the two groups was significantly different ($p<0.0001$). A two-sided p value for the distribution of HSUR2 binding sites in the 3'UTR of target mRNAs relative to a theoretical median of 0.5 (i.e. midway between the stop codon and the polyA tail) (*Figure 2—figure supplement 1A*) was calculated with the Wilcoxon Signed Rank test (alpha = 0.05) after applying the D'Agostino and Pearson normality test ($p<0.001$; did not pass normality test).

## Bioinformatic analyses

Several applications were run to search for consensus sequence motifs in HSUR2 target sequences. Two sequence sets were prepared from the 171 target genes, the variable length minimum interaction sequences presented in *Supplementary file 2* and a set of 100 bp sequences centered over each minimum sequence. These two sets were run through two tools from the MEME analysis suite (http://meme-suite.org) to identify short ungapped (DREME) or gapped (GLAM2) motifs relative to a background derived by shuffling the input sequences. DREME failed to return any motifs with a default E-value of <0.05. GLAM, as the program is designed, returned gapped motifs but they failed to replicate indicating non-convergence. A third attempt at finding a consensus motif utilized the Oligo-Analysis tool in RSAT (http://rsat.sb-roscoff.fr/index.php ) where 7, 8, or 9mer sequences are extracted from the input sequences and scored for enrichment relative to an organism matched background set of sequences. Both the *C. jacchus* upstream no ORF background and a custom background of 10K randomly selected UTRs were run. The motifs found differed by background and only occurred in at most 41 of 167 target sequences. Thus, a clear target sequence consensus motif is not apparent.

The high confidence list of HSUR2 target mRNAs (Supplementary Table 1) was analyzed using the *G*ene *O*ntology en*ri*chment ana*l*ysis and visua*liza*tion tool (GOrilla) (*Eden et al., 2009*) using gene symbol and default parameters, and compared against the complete list of genes sequenced. Alternatively, HSUR2 target mRNAs were classified based on whether their binding to HSUR2 utilizes the miR-16 binding site (miR-16-independent or -dependent target mRNAs) and each list was analyzed separately with GOrilla as described above. All pathways that showed an enrichment p value < 0.001 (default setting not adjusted for multiple hypothesis testing) are shown in Supplementary Table 4.

## Data availability

Source data for all figures in this article are included in its Supplementary Information. The described RNA-seq data have been deposited in the Gene Expression Omnibus under the accession number GSE125371.

## Acknowledgements

We thank Michelle Hastings, Tim Formosa, Dean Tantin, Jeremy Sanford, Javier Cáceres, Brenda Bass, and Wesley Sundquist for critical commentary. This work was supported by a grant of the National Institutes of Health (RO1-GM118829).

## Additional information

### Funding

| Funder | Grant reference number | Author |
|---|---|---|
| National Institute of General Medical Sciences | RO1-GM118829 | Demián Cazalla |

The funders had no role in study design, data collection and interpretation, or the decision to submit the work for publication.

### Author contributions

Carlos Gorbea, Data curation, Formal analysis, Investigation, Methodology, Writing—review and editing; Tim Mosbruger, David A Nix, Formal analysis, Methodology; Demián Cazalla, Conceptualization, Supervision, Funding acquisition, Investigation, Visualization, Methodology, Writing—original draft, Project administration, Writing—review and editing

### Author ORCIDs

Demián Cazalla (iD) https://orcid.org/0000-0002-9076-3655

### Decision letter and Author response

Decision letter https://doi.org/10.7554/eLife.50530.033
Author response https://doi.org/10.7554/eLife.50530.034

## Additional files

### Supplementary files

• Supplementary file 1. This file contains a list of HSUR2 targets identified by iRICC. Gene name, genomic coordinates, and ratios of HSUR2 sample pulldown versus Control sample pulldown.
DOI: https://doi.org/10.7554/eLife.50530.024

• Supplementary file 2. This file contains the sequences in HSUR2 and in target mRNAs involved in base-pairing, binding energies, patterns of interactions identified by RNAcofold in dot-bracket notation, lengths and location of the binding sites in mRNAs, and overlap with regulatory elements in HSUR2.
DOI: https://doi.org/10.7554/eLife.50530.025

• Supplementary file 3. This table contains the sequences of all oligonucleotides used in this study.
DOI: https://doi.org/10.7554/eLife.50530.026

• Supplementary file 4. This table contains all results from our Gene Ontology (GO)-term enrichment analysis.
DOI: https://doi.org/10.7554/eLife.50530.027

• Supplementary file 5. Key Resources Table.
DOI: https://doi.org/10.7554/eLife.50530.028

• Transparent reporting form
DOI: https://doi.org/10.7554/eLife.50530.029

### Data availability

The described RNA-seq data have been deposited in the Gene Expression Omnibus under the accession number GSE125371.

The following dataset was generated:

| Author(s) | Year | Dataset title | Dataset URL | Database and Identifier |
|---|---|---|---|---|
| Gorbea C, Mosbruger T, Nix D, Cazalla D | 2019 | Viral miRNA adaptor differentially recruits miRNAs to target mRNAs through alternative base-pairing | https://www.ncbi.nlm.nih.gov/geo/query/acc.cgi?acc=GSE125371 | NCBI Gene Expression Omnibus, GSE125371 |

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
