## [Decision Letter]

Thank you for submitting your article "Viral miRNA adaptor differentially recruits miRNAs to target mRNAs through alternative base-pairing" for consideration by *eLife*. Your article has been reviewed by three peer reviewers, and the evaluation has been overseen by a Reviewing Editor and Gisela Storz as the Senior Editor. The following individuals involved in review of your submission have agreed to reveal their identity: Bryan Cullen (Reviewer #1); Xavier Roca (Reviewer #3).

The reviewers have discussed the reviews with one another and the Reviewing Editor has drafted this decision to help you prepare a revised submission.

All three of the reviewers were positive about the work, with reviewers 1 and 3 being most enthusiastic, while reviewer 2 asked that the paper be more focused on the technique used. We believe that some expansion on the technique would be an improvement to the paper, but that the emphasis on the biology is appropriate. Please address the points raised by the reviewers as thoroughly as possible.

Reviewer #1:

In this very interesting manuscript, Gorbea et al. extend their previous work arguing that the HSUR2 RNA of HVS functions as an adaptor to recruit either miR-16 or miR-142 to target mRNAs and inhibit their expression. This earlier paper did not define how HSUR2 interacts with target mRNAs as a region of obvious homology was not present. Here, Gorbea et al. first map the targets for HSUR2 binding using a novel technique they call iRICC and they then show that HSUR2 acts as a flexible adaptor that interacts with different mRNAs using overlapping but distinct regions of HSUR2 to bind to different mRNA targets. The mutational analysis presented beautifully demonstrates that this is the case and moreover shows that mutations that disrupt HSUR2:target mRNA interactions can be rescued by compensatory mutations in other bases on HSUR2 that are predicted to allow entirely new base pairs to form. Moreover, some interactions with mRNAs are predicted to involve regions of HSUR2 that are complementary to miR-16 and, in those cases, miR-142 but not miR-16 is required for inhibition. Finally, they also show that they can reconfigure these RNA:RNA interactions so that now the miR-142 complimentary region is required for target mRNA binding while the miR-16 region is now free to recruit miR-16 and in that case the miRNA requirement for inhibition is reversed.

Without going into more detail about the results presented, I will say that I found the data presented uniformly convincing and the conclusions drawn logical and compelling. This paper will certainly be of wide interest to virologists and RNA biologists.

Specific comments:

While I have no technical criticisms, I did have one question. If the region of HSUR2 that binds to target mRNAs includes, for example, the sequence complementary to miR-16, then why doesn't miR-16 compete for binding? If this is a function of relative concentration you might expect the miRNA to be dominant. Alternatively, is there so much HSUR2 that all RNA target sites on both mRNAs and miRNAs are bound? I think it would be interesting to overexpress miR-16 to see if, in this circumstance, mRNA inhibition by HSUR2 via recruitment of miR-142, due to binding of HSUR2 to the mRNA via the miR-16 complementarity region, is selectively inhibited. Otherwise, I have no suggestions for improvement of this very interesting manuscript.

Reviewer #2:

Gorbea et al. describe a novel method (iRICC) for the identification and mapping of RNA:RNA interactions of a specific RNA target in vivo. Such a method is currently lacking and iRICC should be useful to apply to a broad range of RNAs in the future. Here, iRICC is used to define the RNA interactome of the herpesviral Sm-type snRNA HSUR2, which functions as a viral adaptor between mRNAs and the miR-16- and/or miR-142-3p-bound RISC complex. Results show variable modes of HSUR2:target interaction, which is interesting in terms of RNA biology and herpesviral ncRNAs. Data and analyses are of high quality.

1) The authors should discuss whether and how much the reactivity preferences of AMT could bias the results and if there are ways to address such bias.

2) Could the authors discuss potential reasons why the overlap between the results from RICC-seq (Gorbea et al., 2017) and iRICC-seq (this study) is so limited, with only 3 mRNAs in common?

3) iRICC-seq uses a polyA enrichment step (please show in Figure 1). This assumes that HSUR2 mostly targets polyA+ RNAs. HSUR2 exhibits a largely nuclear localization (Chou et al., 1995), while miRISC is normally cytoplasmic, please discuss. Is it possible that non polyA+ targets were missed, which could explain the relatively low number of sites? How critical is the polyA enrichment step? These points should be discussed.

4) The authors should discuss whether failure to identify a statistically enriched sequence motif in the HSUR2 binding sites could be due to small sample size (n=171 vs. thousands usually used for motif analysis).

5) Could the authors perform pathway analysis of the mRNAs that have blocked miR-16 BS vs. accessible miR-16 BS so they can discuss potential relevance of miR-16-dependent sites?

Reviewer #3:

This excellent manuscript reports a strong mechanistic progress towards understanding the role of the viral noncoding RNA HSUR2 in regulating gene expression via recruitment of endogenous microRNAs to target mRNAs. This study is a follow-up of the previous work of the corresponding author describing the characterization of HSUR as viral RNAs together with endogenous proteins assembled into small nuclear ribonucleoprotein particles, which regulate the latent stage of monkey herpes viruses in T cells. HSUR2 was found to bind target mRNAs and then two microRNAs, thus bridging them to the corresponding mRNAs to repress their expression. As only few targets were initially known, authors improved an RNA pulldown technique with HSUR2, which by virtue of crosslink and selective and extensive degradation steps, specifically enriches for mRNAs bound by HSUR2. This elaborate methodology is able to map the crosslinked site at a single-nucleotide resolution, and revealed 171 binding sites in 110 target genes. Authors validated a few of them, however with an impressive 100% validation rate, emphasizing the high power of the technique and the value of the resulting data. Authors used RNA structure predictions to infer the binding mode of HSUR2 to their targets, to surprisingly reveal very different base-pairing arrangements involving different and disconnected segments of the HSUR2 single-stranded 5' end portion. As this segment also binds miR-142-3p and perhaps miR-16, authors nicely used luciferase reporters to firmly establish the flexible base-pairing arrangements and miR dependency. To me, this nice work involves a new technique for high-resolution mapping of RNA-RNA interactions which could be used for other RNAs, together with significant mechanistic insight into a fascinating noncoding RNA. Hence I recommend this manuscript for publication in *eLife*, and I feel that addressing the small comments below would only make this report stronger.

1) Authors identified as many as 171 target mRNAs for HSUR2 in 110 genes, and presumably the viral modulation of these targets altogether helps establish latency. To gain a holistic insight, authors could run Gene Set Enrichment tools or Gene Ontology to see which cellular functions are enriched in this set of genes (apoptosis, cell cycle, etc.). This is important, and could make up one new figure panel and one paragraph description.

2) The new methodology termed iRICC seems applicable for many other RNA-RNA interactions, and authors could stress this a bit more. Importantly, while all tested targets were confirmed, I wonder if or how many targets were missed (in other words, false negatives). Authors should discuss their opinion on how many targets would have been missed, if any.

3) Luciferase reporters show some targets whose HSUR2 regulation depends on both miR-142-3p and miR-16, and inhibition of either one abrogates the repressive effect of HSUR2. This is difficult to conceive for me, because repression of one microRNA is not expected to affect the other – are authors implying that both miRNAs must act together or synergistically in these cases? At the very least this should be discussed.

---

## [Author Response]

Reviewer #1:[…] Specific comments:While I have no technical criticisms, I did have one question. If the region of HSUR2 that binds to target mRNAs includes, for example, the sequence complementary to miR-16, then why doesn't miR-16 compete for binding? If this is a function of relative concentration you might expect the miRNA to be dominant. Alternatively, is there so much HSUR2 that all RNA target sites on both mRNAs and miRNAs are bound? I think it would be interesting to overexpress miR-16 to see if, in this circumstance, mRNA inhibition by HSUR2 via recruitment of miR-142, due to binding of HSUR2 to the mRNA via the miR-16 complementarity region, is selectively inhibited. Otherwise, I have no suggestions for improvement of this very interesting manuscript.

We thank the reviewer for asking this very interesting and important question. The question of how HSUR2 utilizes miR-142-3p and miR-16 to repress target mRNAs is a puzzling one that we have not been able to explain using simple models. Our results show that HSUR2 does not use these two miRNAs in an additive manner for any of the targets tested so far, suggesting that either HSUR2 does not recruit both miRNAs at the same time to a target mRNA (making miR142-3 and miR-16 mutually exclusive for HSUR2-mediated mRNA repression), or that recruiting one miRNA already saturates the repressive capacity of HSUR2. A “mutually exclusive” model for the use of miR-142-3p and miR-16 cannot explain our observation that miR-142-3p activity is

absolutely required for HSUR2-mediated repression for of all targets tested (approximately 25 so far). It is for this that we feel that overexpression experiments as the one suggested by the reviewer will be very hard to interpret and that is why we have so far been able to test only the requirement of each miRNA for HSUR2-mediated mRNA repression. Regarding the first hypothesis proposed by the reviewer, we speculate that since in many cases the regions of

HSUR2 that bind to target mRNAs are extensive, it is possible that miR-16 competes effectively with the target mRNA for the UGCUGCU sequence (miR-16 binding site) present in HSUR2 without effectively disrupting the HSUR2-mRNA interaction. Regarding the second alternative proposed by the reviewer, we think it unlikely since we have already shown that HSUR2 is less abundant than miR-16 in infected cells (Gorbea et al., 2017) and miR-16 activity is not

affected by HSUR2 in infected cells (Cazalla et al., 2010), suggesting that only a small fraction of miR-16 is bound by HSUR2 in infected cells. Further experimentation (currently undergoing) is required to dissect the molecular mechanism by which HSUR2 utilize these two miRNAs to repress target mRNAs.

Reviewer #2:[…] 1) The authors should discuss whether and how much the reactivity preferences of AMT could bias the results and if there are ways to address such bias.

We agree with the reviewer that this is an important point that needs to be discussed in the manuscript. We now mention this in the Discussion section (subsection “iRICC identifies RNA-RNA interactions in vivo”).

2) Could the authors discuss potential reasons why the overlap between the results from RICC-seq (Gorbea et al., 2017) and iRICC-seq (this study) is so limited, with only 3 mRNAs in common?

We now discuss this in the Discussion section (subsection “iRICC identifies RNA-RNA interactions in vivo”). We consider that both methods underestimate the number of targets for HSUR2, and there are substantial methodological differences between them that could account for the limited overlap between the two lists of targets. If we do relax our criteria for calling targets in iRICC, the overlap between the two lists increases substantially. For example, if we do consider genes that are enriched in two out of three iRICC experiments, we have 20 genes that appear in the list of targets identified by RICC-seq (data not shown). We prefer to maintain a stringent criterion to call targets with high confidence rather than relaxing it to increase the number of targets in the lists generated. We believe that our criteria explain the 100% validation rate we have had so far with both methods.

3) iRICC-seq uses a polyA enrichment step (please show in Figure 1). This assumes that HSUR2 mostly targets polyA+ RNAs. HSUR2 exhibits a largely nuclear localization (Chou et al., 1995), while miRISC is normally cytoplasmic, please discuss. Is it possible that non polyA+ targets were missed, which could explain the relatively low number of sites? How critical is the polyA enrichment step? These points should be discussed.

Thanks for the reviewer for these suggestions. The points are now discussed in the subsection “iRICC identifies RNA-RNA interactions in vivo”.

4) The authors should discuss whether failure to identify a statistically enriched sequence motif in the HSUR2 binding sites could be due to small sample size (n=171 vs. thousands usually used for motif analysis).

We agree with the reviewer that this is a possibility and that had we had a larger sample of target sequences we would have probably found some motifs enriched in the target sequences. We think, however, that this does not contradict our conclusions that HSUR2 operates through a flexible mode of base-pairing and that does not use a seed-based mechanism for target recognition.

5) Could the authors perform pathway analysis of the mRNAs that have blocked miR-16 BS vs. accessible miR-16 BS so they can discuss potential relevance of miR-16-dependent sites?

We thank very much the reviewer for this suggestion. We had performed Gene Ontology (GO)-term enrichment analysis. This data is presented in the newly added Figure 2—figure supplement 1D and Supplementary file 4, and the relevance of the new data is discussed in the last paragraph of the subsection “iRICC identifies RNA-RNA interactions in vivo” and subsection “HSUR2 differentially recruits miR-16 to a subset of target mRNAs”.

Reviewer #3:[…] 1) Authors identified as many as 171 target mRNAs for HSUR2 in 110 genes, and presumably the viral modulation of these targets altogether helps establish latency. To gain a holistic insight, authors could run Gene Set Enrichment tools or Gene Ontology to see which cellular functions are enriched in this set of genes (apoptosis, cell cycle, etc.). This is important, and could make up one new figure panel and one paragraph description.

We have followed the reviewer’s suggestion and performed GO-term enrichment (please see response to point 5 from reviewer 2 above).

2) The new methodology termed iRICC seems applicable for many other RNA-RNA interactions, and authors could stress this a bit more. Importantly, while all tested targets were confirmed, I wonder if or how many targets were missed (in other words, false negatives). Authors should discuss their opinion on how many targets would have been missed, if any.

We appreciate the reviewer’s suggestion. We have now discussed this issue in the Discussion section (please see response to points 1 through 3 from reviewer 2 above).

3) Luciferase reporters show some targets whose HSUR2 regulation depends on both miR-142-3p and miR-16, and inhibition of either one abrogates the repressive effect of HSUR2. This is difficult to conceive for me, because repression of one microRNA is not expected to affect the other – are authors implying that both miRNAs must act together or synergistically in these cases? At the very least this should be discussed.

We share the reviewer’s interest in understanding how HSUR2 utilizes these two miRNAs to repress target mRNAs.